# Investigation on Microstructural Evolution and Properties of an Al-Cu-Li Alloy with Mg and Zn Microalloying during Homogenization

**Hongying Li [1,2,3,\*], Weichen Yu [1], Xiaoyu Wang [1], Rong Du [4] and Wen You [4]**

[1]  School of Materials Science and Engineering, Central South University, Changsha 410083, China; bnbyyy_1991@126.com (W.Y.); zhaofei0508@csu.edu.cn (X.W.)

[2]  State Key Laboratory on Lightweight High-strength Structural Material, Central South University, Changsha 410083, China

[3]  Nonferrous Metal Oriented Advanced Structural Materials and Manufacturing Cooperative Innovation Center, Central South University, Changsha 410083, China

[4]  Department of scientific research and development, Southwest Aluminum (Group) Co., Ltd. Chongqing 401326, China; dur99@126.com (R.D.); jpb112@163.com (W.Y.)

\*  Correspondence: lhying@csu.edu.cn; Tel.: +86-731-8887-6692

**Abstract:** The microstructural evolution and properties of an Al-Cu-Li alloy with Mg and Zn microalloying (Al-3.5Cu-1.5Li-0.5Mg-0.4Zn-0.3Mn-0.12Zr-0.06Ti) ingot subjected to homogenization (second-step annealing at 500 °C for 24 h following first-step annealing at 400 °C for 8 h) were investigated. Mg-Zn atom clusters were enriched at the end of dendrites as well as low-melting eutectic phases such as $S$ ($Al_2CuMg$), $T_2$ ($Al_6CuLi_3$), $T_B$ ($Al_{7.5}Cu_4Li$) and $T_1$ ($Al_2CuLi$) in the as-cast alloy. During homogenization, Mg-Zn atom clusters diffused from the segregation to the vacancies, leading to the dissolution of the low-melting eutectic phases. Not only $Al_3Zr$ particles were observed at 500 °C, but more fine and uniform spherical dispersoids appeared, which were assumed as $Al_3(Zr_xTi_yLi_{1-x-y})$. Mg and Zn microalloying can promoted the nucleation of $Al_3Zr$ and $Al_3(Zr_xTi_yLi_{1-x-y})$ dispersoids, as well as $T$ ($Al_{20}Cu_2Mn_3$) phases, which all inhibited recrystallization effectively and improve the uniformity of the grains due to the strong pinning effect. The yield ratio was decreased from 0.81 to 0.52 with the yield strength decreased from 172 MPa to 61 MPa, which showed better plastic deformation ability of the alloy subjected to homogenization. In addition, the dissolution of low-melting eutectic phases and formation of $Al_3(Zr_xTi_yLi_{1-x-y})$ dispersoids resulted in the significant improvement on thermal stability.

**Keywords:** Al-Cu-Li alloy; homogenization; microstructures; properties; yield ratio; thermal stability

## 1. Introduction

At present, Al-Li alloys are increasingly being studied as the structural materials with the most potential for development in the 21st century. The third-generation Al-Li alloys are widely used in military and aerospace applications because of their superior properties such as low density, high specific strength as well as damage tolerance [1,2].

With reference to the Aluminum Association, it is generally believed that the third-generation Al-Li alloys can be divided into high-strength weldable Al-Li alloys containing Sc (such as 1445 and 1460) [3–5], high-plasticity Al-Li alloys containing Ag (such as 2195 and 2198) [6,7] and low-cost Al-Li alloys containing Zn (such as 2060, 2099 and 2A97) [8–10]. As a key technology for U.S. Low Cost Manufacturing System, the cost of Al-Cu-Li alloys microalloying Mg and Zn is lower than that of rare earths and their recyclability is higher due to the addition of Zn and Mn elements [11]. The addition

of Mg and Zn to 7XXX series alloys can lead to the formation of $\eta$ (MgZn$_2$) phase during heat treatment [12,13], besides, the addition of Zn can also promote the precipitation of $T_1$ phase as Ag-like effect in 2198 alloy [14,15]. However so far, the specific mechanism of the influence of Zn on the microstructure and properties of the alloy during solidification, deformation and heat treatment is still unclear.

It is still more common to prepare Al-Li alloys by melting and casting at the factory. Due to the active chemical property of Li, the degree of constitutional supercooling in the solidified alloy is high, which causes serious segregation. The homogenization process heated to a certain temperature is usually used to eliminate the segregation and promote the thermal workability of the alloy. Considering the diffusion of one component in an ideal binary solution during homogenization, Fick's first law in this case may be written as Equation (1):

$$J = -D\frac{\partial c}{\partial x} \tag{1}$$

where the diffusion flux per unit area perpendicular to the diffusion direction is proportional to concentration gradient at the cross section per unit time. The relationship between diffusion coefficient $D$ and annealing temperature can be expressed by Arrhenius equation as Equation (2):

$$D = D_0 exp\left(-\frac{Q}{RT}\right) \tag{2}$$

where it can be seen that as long as the temperature is slightly increased, the diffusion process will be greatly accelerated. However, this is not exactly the case. If the heat treatment temperature is too high, liquid phases will be generated and the alloy might be overburnt. Therefore, the step homogenization process is even more pronounced in Al-Cu-Li alloys considering the active chemical property of Li and the presence of many Li-containing low-melting eutectic phases [16]. It has been reported that the formation of the *W* phase (AlCuSc) due to the interdiffusion of Cu and Sc during two-step homogenization was investigated in 1469 alloy with the high Cu content and Sc addition [17]. However, it has rarely been reported to study the microstructures and properties of Al-Cu-Li alloy microalloying Mg and Zn during the homogenization. The diffusion mechanism and specific role of Zn in the homogenization process are not yet clear. In addition, the Mg-Zn atom clusters were detected in many aging Al-Cu-Li-Mg-Zn alloys, which displayed similar characteristics to the Mg-Ag atom clusters observed in the Al-Cu-Li-Mg-Ag alloys [18–20]. It is possible to form the enrichment of Mg-Zn atom clusters in the alloy during solidification as well. Moreover, the vacancy bonding energy of Mg-Zn atom clusters is very high, so the Mg-Zn atom clusters are likely to capture the free atoms during homogenization, leading to promotion on the precipitation behavior, which has not been experimentally proven.

Consequently, the aim of the present study was to investigate the features of segregation in as-cast alloys and the evolution of phases during homogenization by electron probe microanalysis and transmission electron microscopy. The mechanical properties and thermal stability were also detected in the hope of better plastic deformation ability for the following thermal deformation of the alloy.

## 2. Materials and Methods

The rectangular ingot (220 mm × 140 mm × 35 mm) from the factory was detected for the present researches, whose chemistry was shown in Table 1. The cube samples (15 mm × 15 mm × 1.5 mm) cut from the central part of the ingot were subjected to a two-step homogenization in salt bath furnace, in which the samples were heated at 400 °C for 8 h followed by 500 °C for 24 h and cooled by water quench. The schematic of the thermal history for the homogenization was shown in Figure 1, in which highlights (with dots) were the times of samples taken from the furnace to determine the evolution of microstructure.

**Table 1.** The chemistry of Al-Cu-Li-Mg-Zn alloy (wt.%).

| Cu | Li | Mg | Zn | Mn | Zr | Ti | Fe | Si | Al |
|-----|-----|-----|-----|-----|------|------|--------|--------|-----|
| 3.5 | 1.5 | 0.5 | 0.4 | 0.3 | 0.12 | 0.06 | ≤0.10 | ≤0.08 | Bal |

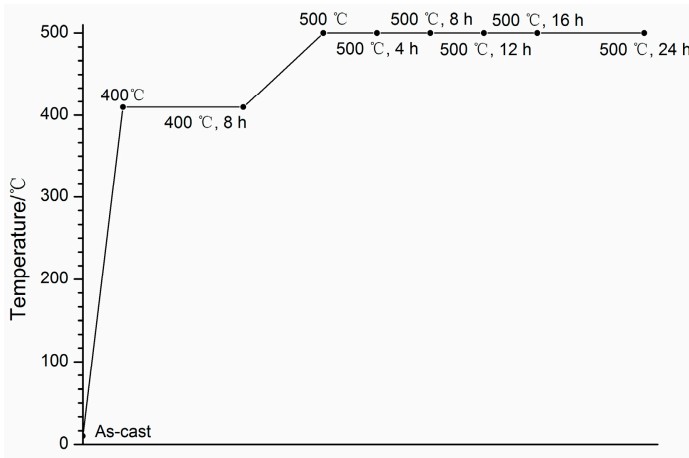

**Figure 1.** Schematic of the thermal history used for the homogenization.

Microstructure characterizations before and after homogenization of the samples were carried out by using Leica DMI3000 polarizing microscope (Leica, Wetzlar, Germany). The samples were subjected to electrolytic polishing by $HBF_3$ (16.8 g/L). Phase constitution and evolution during homogenization were detected by scanning electron microscope (SEM) Quanta-200 (FEI, Hillsboro, OR, USA) with energy dispersive X-ray Spectroscopy (EDS) GENESIS 60S (EDAX, Mahwah, NJ, USA) with the resolution of 131 ev. A precision around ±1% was obtained owing to factors such as uncertainties in the compositions of the standards and errors in the various corrections which need to be applied to the raw data. Segregation and diffusion of alloying elements were characterized by JEOL JXA-8230 electron probe microanalysis (EPMA) instrument (JEOL, Tokyo, Japan) with the acceleration voltage of 20 KV. The observation of precipitation was achieved by Tecnai-$G^2$ 20 transmission electron microscope (TEM, FEI, Hillsboro, OR, USA) with the acceleration voltage of 200 KV. The high-angle annular dark field (HAADF) image was investigated on the distribution of elements on the precipitation by Titan $G^2$ 60-300 (FEI, Hillsboro, OR, USA). The samples were polished by twin jet electropolishing system with the electrolyte of mixture of 25% $HNO_3$ + 75% $CH_3OH$ (volume fraction). The voltage and the electric current were set at 15 V and 10–20 mA respectively. The temperature was controlled below −35 °C during the operation. Sigma 2008 eddy current conductivity meters (Xiamen Tianyan, Xiamen, China) was used in measuring the electrical conductivity of the alloy at room temperature of 20 °C. The mechanical properties were measuring by a universal testing machine MTS 858 (MTS, Eden Prairie, MN, USA) with a strain rate of $1.0 \times 10^{-3}$ s$^{-1}$. The samples (Figure 2) were cut from the ingots and homogenized under various conditions before the tensile test. Both results took the average of 5 samples.

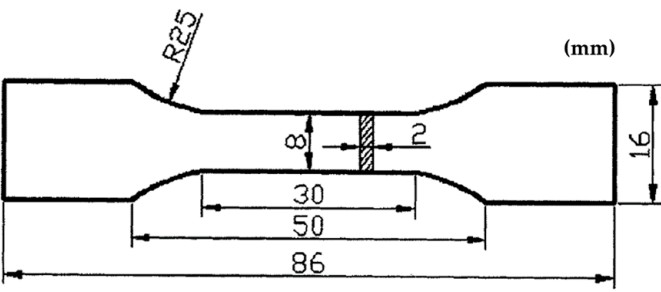

**Figure 2.** Tensile test sample of the alloy.

## 3. Results

### 3.1. Optical Microstructure

The optical microstructures with cross-polarized light on grains and boundaries of the alloy before and after homogenization were shown in Figure 3. There existed typical dendrites in the as-cast alloy in Figure 3a obviously. Precise grain size can be obtained by EBSD according to the previous work published by Riestra [21]. In the optical micrograph, the average size of grains was estimated roughly to be 350–450 µm. A great number of micro-segregations were formed within the grains. The grain boundaries and dendrites consisted of coarse low-melting eutectic phases. After homogenization, as shown in Figure 3b, dendritic grains and micro-segregation have been eliminated and most of coarse eutectic phases were disappeared. The grain boundaries became sharp and smooth.

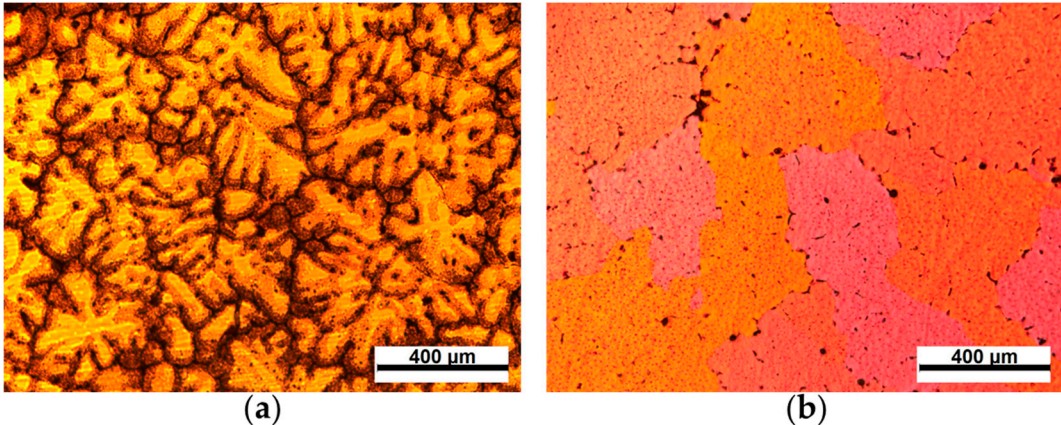

**Figure 3.** Optical microstructures of the (**a**) as-cast and (**b**) homogenized alloy with cross-polarized light.

### 3.2. Segregation Evolution

The elements distribution of Cu, Mg and Zn was investigated by EPMA in the as-cast, first-step homogenized (400 °C, 8 h) and two-step homogenized (400 °C, 8 h + 500 °C, 24 h) alloys in Figure 4, where the concentration of alloying elements in different regions was listed next to the Figure. The non-uniform distribution of Cu, Mg and Zn was obvious in the as-cast alloy. Cu had the highest concentration up to 40.35% in the coarse continuous phases. The Mg and Zn tended to be concentrated together on and around the dendrites. After the first-step homogenization, the color of Cu in the coarse continuous phases had merely changed comparatively in Figure 4a,b, which indicated that the diffusion of Cu was not clear at 400 °C. The segregation of Mg around the dendrites still existed in Figure 4e. But many bright particles represented for phases with Mg enrichment were visible in the grains, indicating that Mg atoms were enriched at the end of the phases near the dendrites. The distribution of Zn was relatively uniform in the grains after 400 °C shown in Figure 4g. After all, the segregation of Mg and Zn formed around the dendrites began to be eliminated. After homogenized, the color of Cu, Mg and Zn in the grains became more uniform as Figure 4c, f and i. No dark color area was found in grains. The two-step homogenization treatment successfully eliminated the segregation of Cu, Mg and Zn.

Note that the color of Cu, Mg and Zn in the grains changed during the homogenization process. The color of Al matrix became light after homogenization, which displayed blue in Figure 4c of Cu and green in Figure 4f of Mg. This result indicated a rise of Cu and Mg content in the matrix. In contrast, the concentration of Zn in the grains decreased in Figure 4i. The results revealed that Cu and Mg atoms tended to move from the coarse phases on the dendrites towards the Al matrix during homogenization treatment, while the movement of Zn atoms was opposite, from the Al matrix towards the second phases.

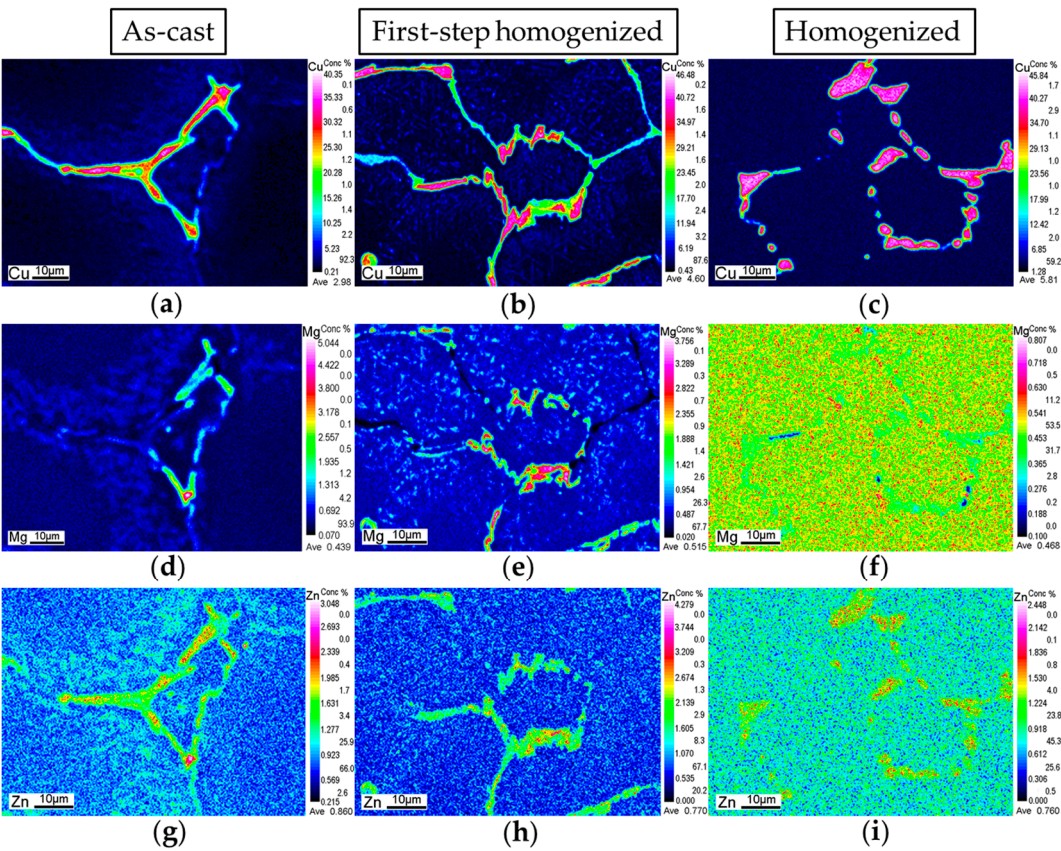

**Figure 4.** Elements mapping of the selected regions in as-cast, first-step homogenized (400 °C, 8 h) and second-step homogenized (400 °C, 8 h + 500 °C, 24 h) alloys by EPMA: (**a–c**) Cu, (**d–f**) Mg and (**g–i**) Zn.

Further analysis of elements distribution after homogenization was detected in Figure 5. The distribution of alloying elements had obvious regional characteristics, indicating that the non-equilibrium structures of the as-cast alloy had been transformed into the equilibrium phases. Three different distributions of Cu concentration were observed obviously on the grain boundary in Figure 5a, which were divided into area A, B, and C. The element concentration in areas A–C can be obtained in Table 2. The average concentration of Cu atoms was the highest (45.6%) at area A, where the concentrations of Mg and Zn were also relatively highest. No enrichment of Zr, Mn and Fe were observed at area A. The average concentration of Cu atoms at point B was 27.5%, where the enrichment of Mg and Zn were apparent with lower concentration than that in area A. The enrichment of Mn and Fe was found in area B, where the color was obviously brighter than the surrounding in Figure 5e,f. It is worth noting that there was enrichment of Zr in bright color with the average concentration of ~0.5% at area B, which was pointed by yellow arrow in Figure 5d. Besides, the color of Fe in the particle pointed by the arrow was also different from the surrounding in Figure 5f, indicating that the concentration of Fe was higher here. The average concentration of Cu atoms in area C was 20.9%, where Mg was hardly present because the color of Mg was the darkest in Figure 5b. The existence of Zn atoms was observed but the enrichment was not formed because the color of Zn at area C was very uniform with the grain. The Mn and Fe atoms were enriched in area C and have performed the highest average concentration relatively. Moreover, unlike Cu, Mg, Zn, Zr and Fe, the distribution of Mn atoms in the grain was not uniform in Figure 5e. According to previous work, the toughness phase, $T$ ($Al_{20}Cu_2Mn_3$) phase [22], might formed in grains and promoted the solute free zone along the grain boundary in darker color shown in Figure 5e.

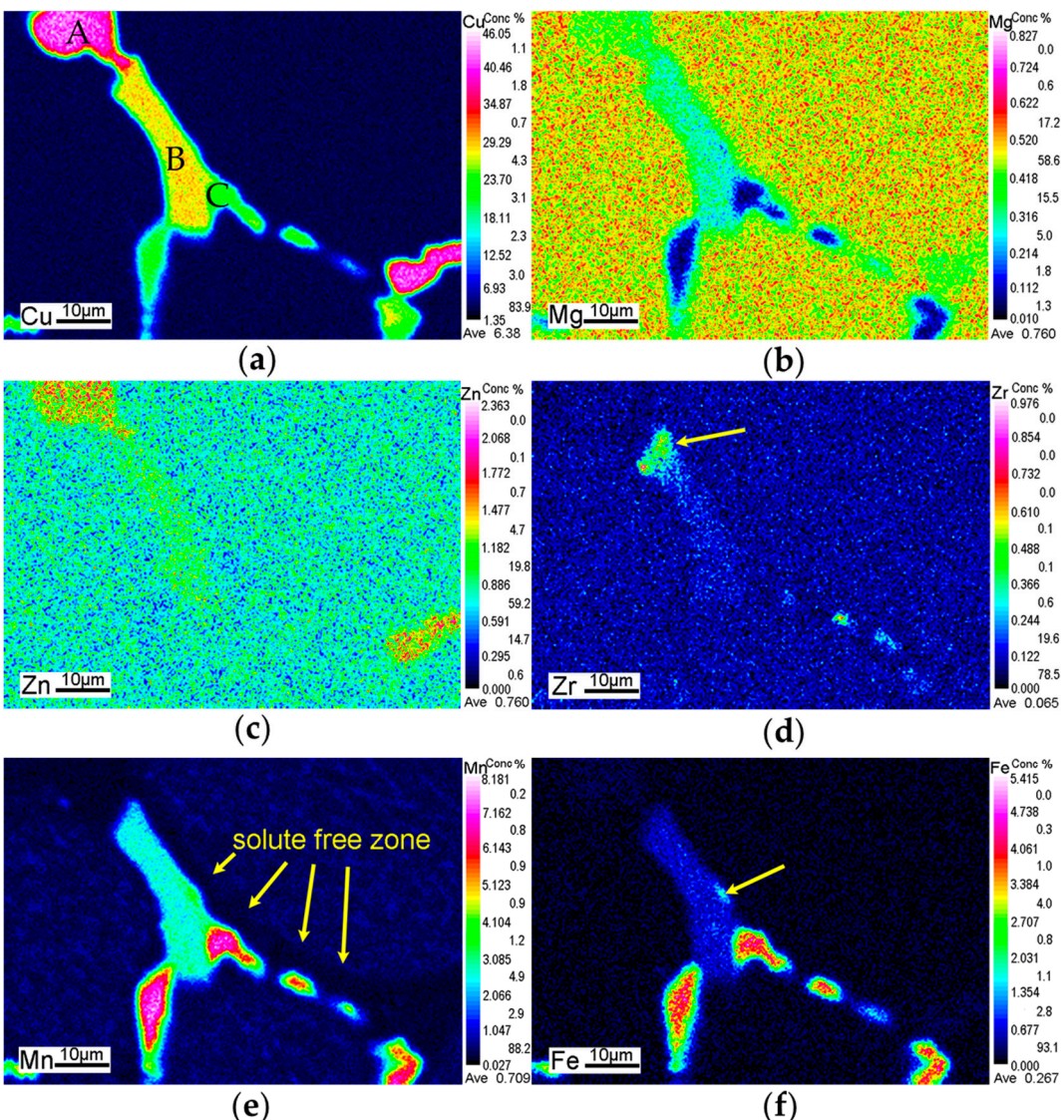

**Figure 5.** Elements mapping of the as-cast alloy by electron probe microanalysis (EPMA): (**a**) Cu, (**b**) Mg, (**c**) Zn, (**d**) Zr, (**e**) Mn, and (**f**) Fe.

**Table 2.** Element concentration in areas A–C in Figure 5 (%).

| Area | Cu | Mg | Zn | Mn | Zr | Fe |
|------|------|-----|-----|-----|------|-----|
| A | 45.6 | 0.5 | 2.2 | - | - | - |
| B | 27.5 | 0.4 | 1.2 | - | ~0.5 | - |
| C | 20.9 | - | 0.7 | 7.5 | - | 4.6 |

*3.3. Constituent Particles*

The intermetallic constituent particles were characterized by backscattered scanning electron microscope (Quanta-200, FEI, Hillsboro, OR, USA) in Figure 6, where the chemical compositions were detected by EDS shown in Table 3. The dendrites and grain boundaries consisted of continuous coarse phases and small particles with higher contrast in the as-cast alloy in Figure 6a. A great number of significant rod-like phases with the size of 12–20 µm were also visible around the dendrites. Six different phases were observed in and around the dendrites and grain boundaries, which were numbered with A–F according to their different sizes, appearances and contrasts. As shown in point A, the atomic ratio of Al and Cu atoms in the long and coarse phase is close to 2:1, which was formed

almost on the dendrites and grain boundaries. The closest phase might be determined to be $\theta$ phase ($Al_2Cu$) [23]. The atomic ratio of Cu and Mg atoms in the gray eutectic phase of point B is close to 1:1, therefore the closest phase might be $S$ phase ($Al_2CuMg$) [24]. In the bright phase with a larger contrast of point C, EDS analysis revealed a high content of Cu, where the atomic ratio of Al and Cu atoms is close to 7:4. Because lithium is too light to be detected, the phase of point C might be identified as $T_B$ phase ($Al_{7.5}Cu_4Li$) [16]. In the white particle of point D on the dendrite, the atomic ratio of Al to Cu atoms is close to 6:1, which might be $T_2$ ($Al_6CuLi_3$) [25]. In the low-contrast coarse phase shown in point E, the Fe and Mn contents are higher than the other phases, which means the existence of AlCuFeMn phase, a refractory impurity phase in Al-Cu alloys [26]. In area F, the atomic ratio of Al and Cu atoms in the rod-like phases is close to 2:1, which is closest to $T_1$ ($Al_2CuLi$) phase judged by the appearance and element composition synthetically [23].

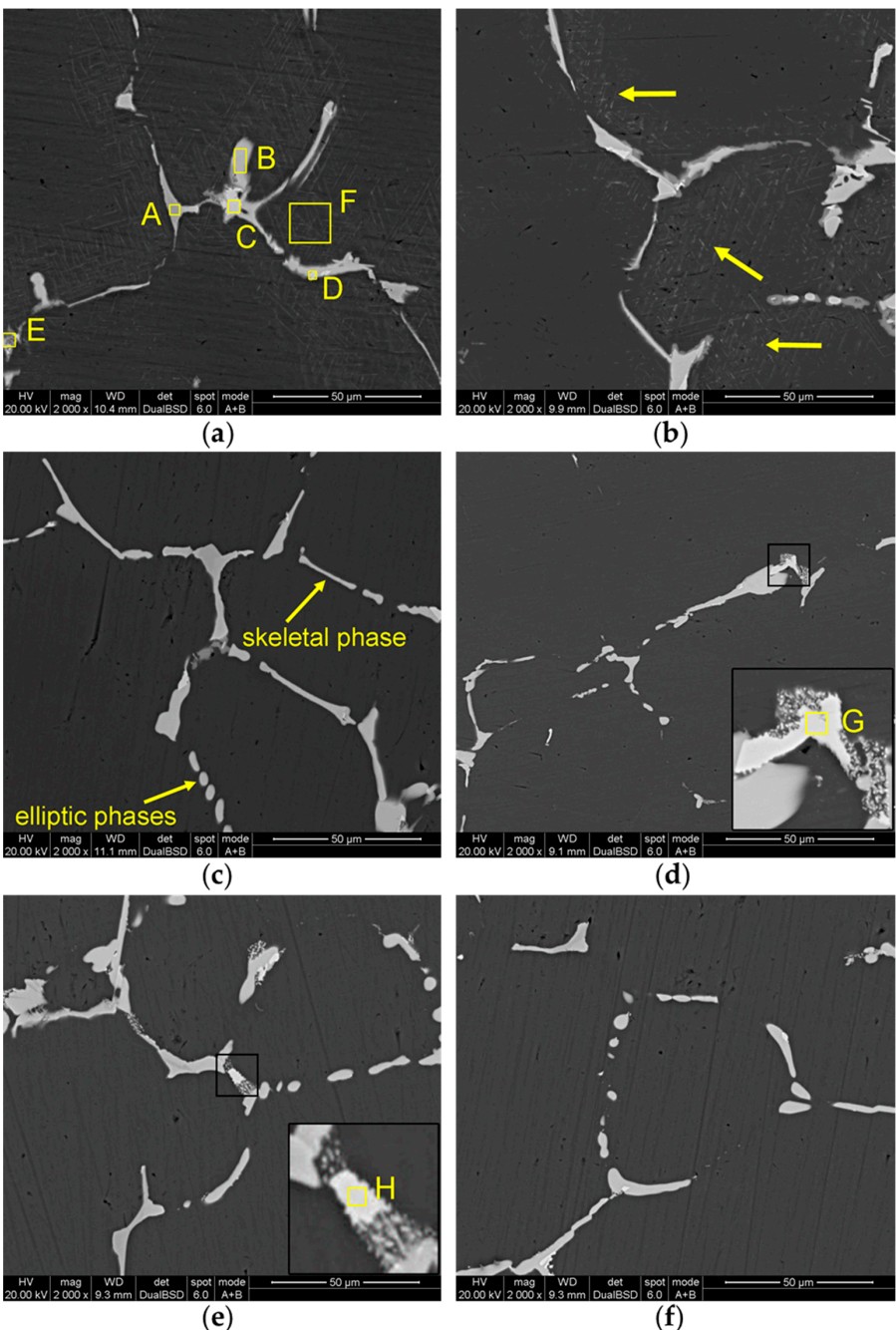

**Figure 6.** *Cont.*

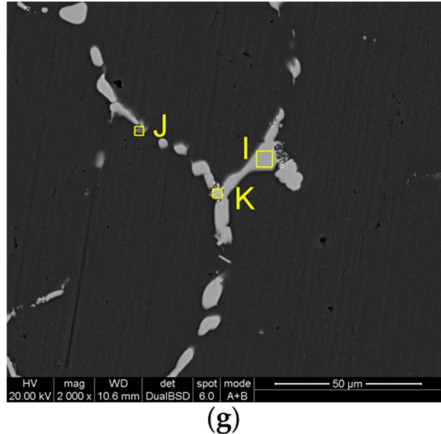

**(g)**

**Figure 6.** SEM images on microstructures of the alloys (**a**) as-cast and homogenized at (**b**) 400 °C for 8 h subsequent to second-step homogenization at 500 °C for (**c**) 4 h, (**d**) 8 h, (**e**) 12 h, (**f**) 16 h and (**g**) 24 h.

**Table 3.** Chemical composition of areas in Figure 6 (at.%).

| Area | Al | Cu | Mg | Zn | Mn | Fe | Zr | Closet Phase |
|------|------|------|------|-----|-----|-----|-------|------------------|
| A | 67.6 | 29.3 | 1.5 | 1.2 | 0.3 | 0.2 | - | $Al_2Cu$ |
| B | 70.1 | 15.1 | 13.6 | 0.9 | 0.2 | 0.2 | - | $Al_2CuMg$ |
| C | 61.7 | 34.6 | 1.4 | 1.3 | 0.4 | 0.6 | - | $Al_{7.5}Cu_4Li$ |
| D | 80.1 | 13.4 | 5.1 | 0.5 | 0.1 | 0.2 | - | $Al_6CuLi_3$ |
| E | 79.7 | 11.3 | 1.6 | 0.4 | 2.4 | 4.6 | - | AlCuFeMn |
| F | 61.4 | 30.7 | 3.3 | 3.2 | 0.5 | 0.7 | - | $Al_2CuLi$ |
| G | 64.8 | 31.8 | 1.4 | 1.0 | 0.1 | 0.2 | ~0.5 | $Al_2Cu$ |
| H | 64.0 | 32.0 | 0.9 | 1.2 | 0.3 | 0.4 | ~0.5 | $Al_2Cu$ |
| I | 65.9 | 30.3 | 1.9 | 1.0 | 0.1 | 0.2 | - | $Al_2Cu$ |
| J | 75.2 | 11.9 | 0.3 | 0.2 | 5.5 | 6.7 | - | AlCuFeMn |
| K | 67.1 | 29.6 | 1.3 | 0.8 | 0.1 | 0.3 | ~0.5 | $Al_2Cu$ |

Heating at 400 °C for 8 h, phases on the dendrites partly began to dissolve and shrink. Moreover, the grain boundaries and dendrites have already become disconnected in Figure 6b while those brighter eutectic phases still exist. The rod-like phases around the dendrites and grain boundaries also began to dissolve, which turned to be smaller both in size and number pointed as the yellow arrow. After homogenizing at 400 °C for 8 h followed by a high temperature step at 500 °C for 4 h in Figure 6c, the coarse phases on the grain boundary were gradually spheroidized. The sharp grain boundaries began to become skeletal and elliptic, and no rod-like phases were observed anymore. Prolonging homogenization time to 8 h and 12 h at 500 °C, a bright irregular phase was observed both in Figure 6d,e, which was partly transforming into small particles gradually. By using EDS analysis in the point G and H, the phase was revealed closely to be a Zr-enriching $Al_2Cu$ non-equilibrium intermetallic phase, where the atomic ratio of Al and Cu was close to 2:1. After homogenized at 500 °C for 16 h, the low-melting eutectic phases were further spheroidized and dissolved, and no white coarse phase was observed on the grain boundary. The microstructure of the alloy that went through the complete homogenization procedure, namely 400 °C/8 h + 500 °C/24 h was shown in Figure 6g. Only a small quantity of coarse phases remained in the alloy as shown in point I, J and K, which were detected to be $Al_2Cu$, AlCuFeMn as well as Zr-enriching $Al_2Cu$ phases, respectively.

### 3.4. Dispersoids

Precipitation of the alloy subjected to two-step homogenization was detected by high-resolution TEM (HRTEM) and high-angle annular dark field-scanning transmission electron microsopy (STEM-HAADF) in Figure 7, where the zone axis of diffraction pattern was along $[110]_{Al}$ in Figure 7a.

The diffraction pattern showed the spots distributed symmetrically at $1/2(1\bar{1}0)_{Al}$ and their equivalent positions, which was represented for $Al_3Zr$ in Al-Li alloy with the zone axis of $[110]_{Al}$. A few spherical phases with size of 20–30 nm were observed in the grain in Figure 7b, which were coherent with the matrix. According to the dark field image in Figure 7c, in addition to the large spherical particles, a large number of tiny bright spots were also observed in the yellow square. Element analysis in the spherical phase with size of 20–30 nm was carried out by HAADF-Mapping in Figure 6d–f, which provided the experimental evidence of Zr atoms enriched in the spherical phase. In summary, the spherical phase was judged as a coherent phase $Al_3Zr$ with FCC-based $L1_2$ structure formed during homogenization at 400 °C preferentially [27]. However, it is worth noting that although the diffraction patterns of $Al_3Zr$ was very bright and sharp at the center, the outer part was weak and the light was divergent, which is a typical characteristic of the diffraction spots formed by plenty of small-sized precipitates. The $Al_3Zr$ phase with size of ~20 nm does not meet with this characteristic. The HRTEM was detected in Figure 7g, showing the tiny phases all over the matrix with size of 1–2 nm pointed as yellow arrows, which might be the tiny bright spots in Figure 7c. These nanoscale precipitates were consistent with the characteristic of diffraction spots, presumably as tiny $Al_3(Zr_xTi_yLi_{1-x-y})$ particles. These uniform and tiny particles provided the alloy better thermal stability due to the strong pinning dislocation ability, which is a very rare phenomenon in alloys microalloying no Mg or Zn. The large number of Mg-Zn atom clusters diffused into the matrix during homogenization may be the key to forming $Al_3(Zr_xTi_yLi_{1-x-y})$ particles.

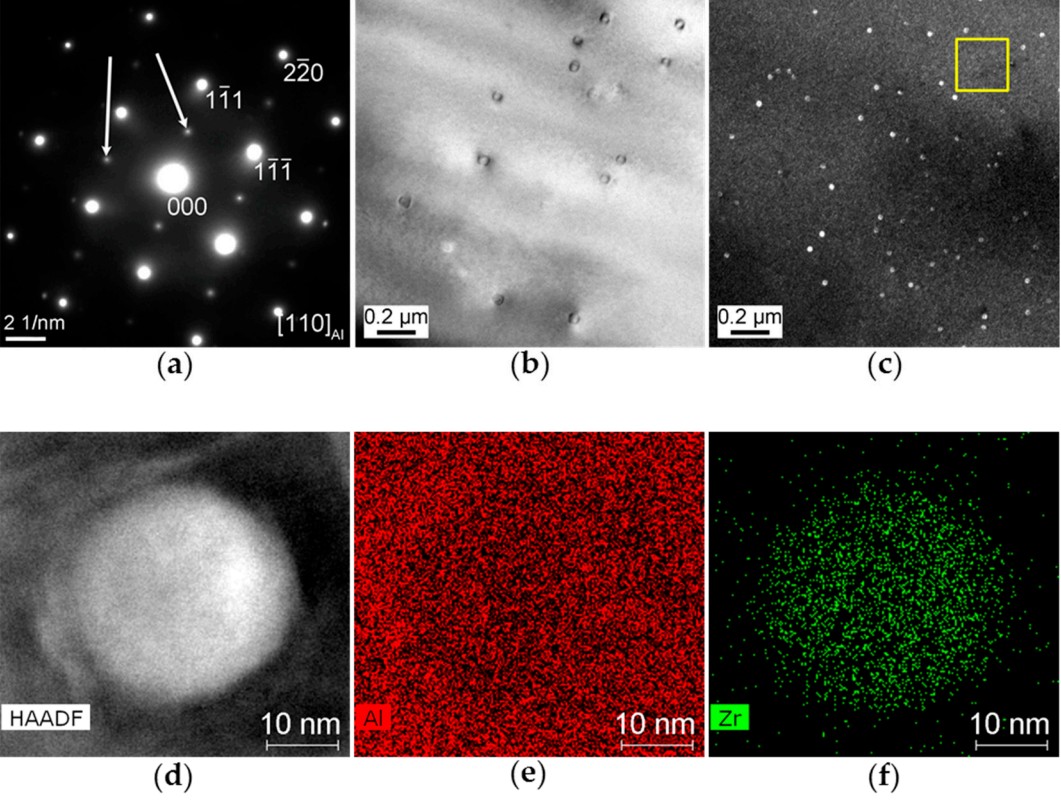

**Figure 7.** *Cont.*

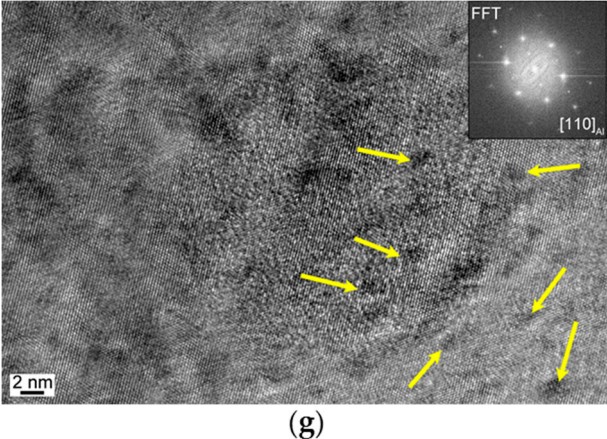

**(g)**

**Figure 7.** TEM images on disperoids with (**a**) diffraction pattern with zone axis of [110]$_{Al}$ through (**b**) bright field and (**c**) dark field, (**d**) STEM-HAADF (high-angle annular dark field-scanning transmission electron microsopy) and elements mapping of (**e**) Al and (**f**) Zr in the particle and (**g**) HRTEM (high-resolution TEM) image of the fine precipitates with FFT.

Figure 8 showed the TEM images and EDS analysis of the micron precipitates in the alloy after homogenization, where Figure 8b–d are the EDS analysis of particle A, B and C in Figure 8a. It can be seen that rod-like precipitates with different contrast and size were observed in the alloy. The EDS analysis shown as Figure 8b indicated that particle A contains Al, Cu and Mn, presumably as *T* phase, which was consistent with the EPMA results in Figure 5. Moreover, particle B was darker in color, which should be the result of the thick contrast, indicating heavy elements in particle B. Then EDS analysis in Figure 8c proved that Zr was enriched in particle B besides Al, Cu, Mg, Mn, Zn and Ti. The possible cause of particle B was the nucleation and growth of the *T* phase on Al$_3$(Zr$_x$Ti$_y$Li$_{1-x-y}$) particles, resulting in a distortion around the composite phases and attracting Mg-Zn atom clusters in the matrix. Interestingly, Zn was detected alone except Mg with Al, Cu and Mn in particle C, which might be caused by solid solution of Zn atoms and *T* phase after high temperature homogenization.

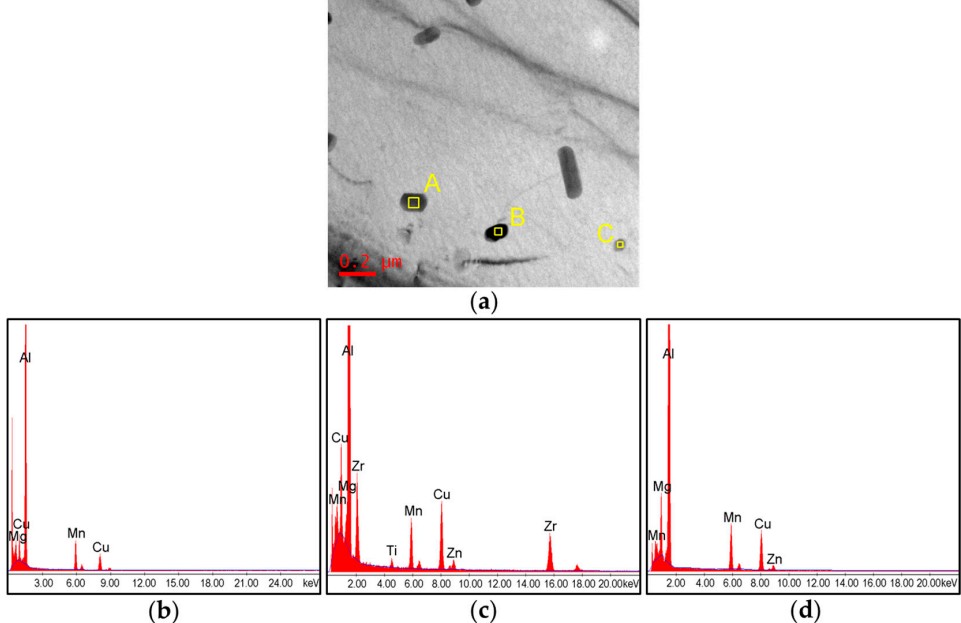

**Figure 8.** TEM images of (**a**) phases in the homogenized alloy and EDS analysis of (**b**) point A, (**c**) point B and (**d**) point C in the rod-like phases.

## 4. Discussion

### 4.1. Microstructural Evolution Analysis

Dendritic grains were observed on the as-cast alloy as shown in Figure 3, which should be the result of composition fluctuations during the non-equilibrium solidification of the alloy [28]. Some intermetallic constituent particles were found formed during solidification of the molten alloy at relatively high temperatures, and their sizes lie in the range of one to several tens of microns. The limited solubility of Li in Al is 4 wt.%, which decreases to 1 wt.% at room temperature. Due to the lightness, Li was difficult to detect by EPMA and EDS, so Li was not found in the elemental analysis of as-cast alloys. However, the Li content in as-cast alloy is 1.3–1.8%, so the supersaturation of Li in the ingot after solidification was very high, which means the easy formation of coarse primary phases. Those large number of rod-like phases around the grain boundaries were presumed as primary phases of $T_1$ as shown in Figures 4 and 6. Furthermore, XRD analysis of the as-cast alloy showed that $T_B$ and $T_2$ were also existed in Figure 9. This is because the chemical property of Li is very active, which resulted in nucleation intermetallic constituent particles with Cu on the grain boundaries during the solidification process. XRD also showed the presence of $Al_6Mn$ in the as-cast alloy. This primary phase is often accompanied by the presence of $Al_6(MnFe)$, which was also not observed in the SEM. It is worth noting that these Li-containing primary phases ($T_1$, $T_B$ and $T_2$) were also enriched with clusters of Mg-Zn atoms.

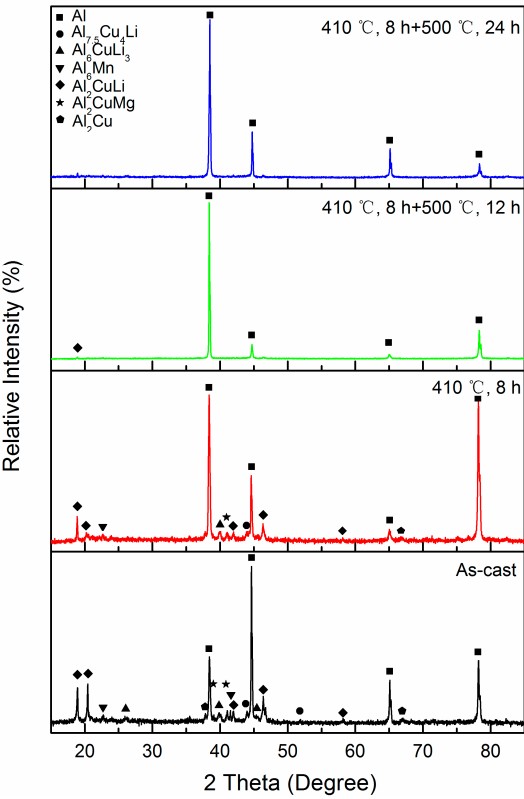

**Figure 9.** XRD analysis of as-cast alloys and alloys subjected to different homogenization.

The limited solubility of Cu in the Al matrix is 5.67 wt.%, which can improve the cutting performance of the alloy as well as certain solid solution and precipitation strengthening. Figure 4 showed that there was a large amount of Cu on the grain boundaries in the as-cast microstructure, forming dendritic segregation. This is because the process of uniform diffusion of Cu atoms in the solid phase occurred slowly, which easily formed coarse intermetallic constituent particles like $Al_2Cu$ on the grain boundaries together with Al atoms [29]. When Mg and Cu were simultaneously added,

$S$ ($Al_2CuMg$) intermetallic constituent phases were easily formed and detected in Figure 6 [30,31]. Mg and Zn were detected to concentrate on almost all over the precipitates and dendrites because Mg and Zn atom clusters were formed during solidification of the as-cast alloy, which adsorbed a large number of free atoms in matrix and promoted the formation of primary phases. Therefore, the dendrites in the as-cast alloys are mainly caused by the non-uniform distribution of components formed by Cu, Mg and Zn. The addition of Mn and Zr will inhibit recrystallization obviously and improve the workability and corrosion resistance of Al-Li alloy [32]. Coarse primary phases with Zr were formed during solidification, meanwhile, Mn was generally detected in low-melting eutectic phases at the dendritic gaps with the impurity Fe [33]. Those coarse insoluble constituent particles have minor effects on strength but adversely affect ductility and fracture toughness, which were easily melted in the subsequent hot deformation process, forming crack initiation sites and resulting in rolling cracks [34].

Based on the investigation results above, the schematic illustration on microstructural evolution during homogenization was displayed on Figure 10. The intermetallic constituent particles form during solidification of the molten alloy at relatively high temperatures, and their sizes lie in the range of one to several tens of microns such as $Al_2Cu$, $Al_2CuMg$ and AlCuFeMn phases. Many kinds of Li-containing eutectic phases with Mg-Zn atom clusters were nucleated in the dendrites, which was surrounded by primary coarse $T_1$ phases. After first-step homogenization, $T_2$, $T_B$, and $T_1$ phases dissolve preferentially. However, only a small amount of $Al_2Cu$ phase dissolved due to the low diffusion rate of Cu. Mg atoms diffused into the matrix from the primary phases and Zn diffused to the Cu-containing phases, accompanied by the combination between Mg-Zn atom clusters and vacancies. After second-step homogenization for 12 h, only Fe-containing AlCuFeMn particles still existed. The $T$ phases and $Al_3(Zr_xTi_yLi_{1-x-y})$ dispersoid particles nucleated at the Mg-Zn atom clusters. Finally, dendritic segregation was completely eliminated by step homogenization. Already formed $T$ phase and $Al_3(Zr_xTi_yLi_{1-x-y})$ particles grew further and fine and uniform $Al_3(Zr_xTi_yLi_{1-x-y})$ particles were newly generated. The alloy showed non-recrystallization characteristic and improvement on hot workability after homogenization.

## 4.2. Properties Evolution Analysis

The mechanical properties and electrical conductivity of the alloy during homogenization were detected in Figure 11, and the value of ultimate tensile strength (UTS), yield strength (YS), yield ratio and electrical conductivity were list in Table 4. The homogenization time of 12 h means homogenization at 400 °C for 8 h following by homogenization at 500 °C for 4 h and so on. As the homogenization proceeds, the UTS of the alloy decreased sharply from 212.5 MPa to 143.5 MPa, while the YS decreased from 172.1 MPa to 101.9 MPa at 400 °C for 8 h as the yield ratio decreased from 0.81 to 0.71. Prolonging homogenization time at 500 °C, the alloy continued to soften as the yield ratio decreased to 0.52. As the homogenization proceeds, the electrical conductivity of the alloy continued to increase from 6.52 MS/m to 8.66 MS/m.

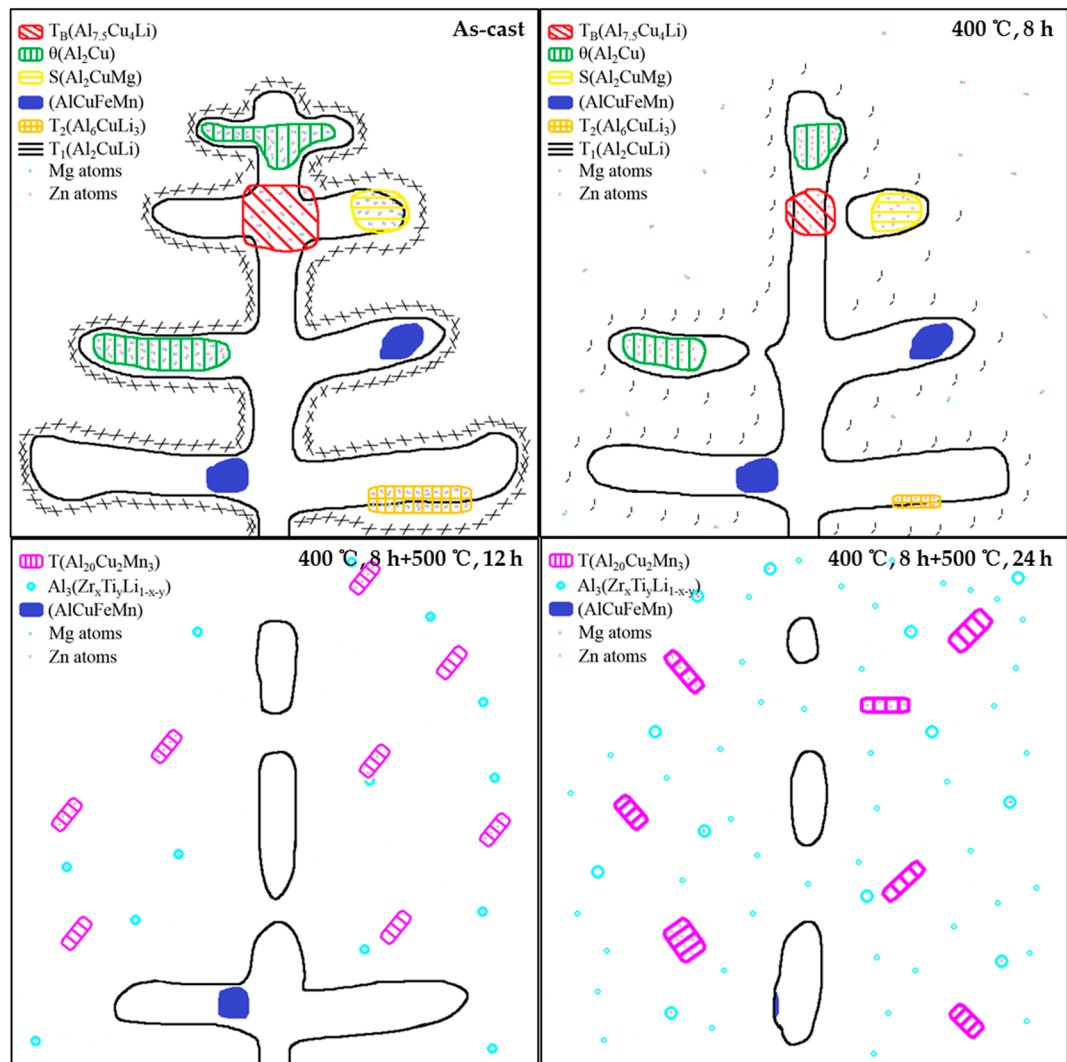

**Figure 10.** Schematic illustration on microstructures of the alloy during homogenization.

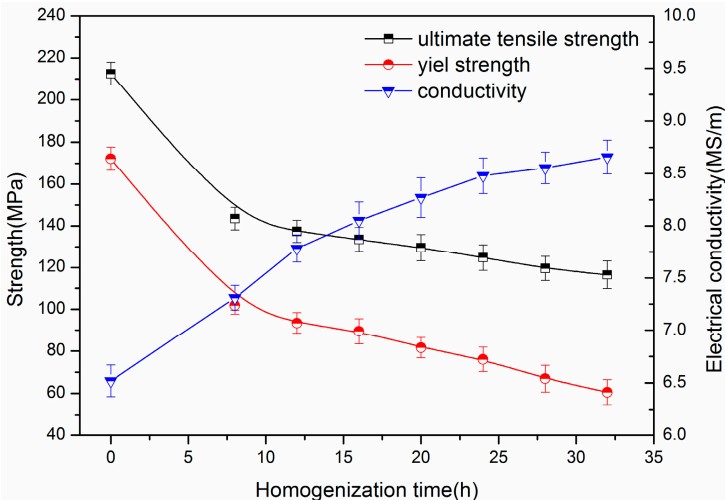

**Figure 11.** Mechanical properties and electrical conductivities of the alloy during homogenization.

**Table 4.** Mechanical properties, yield ratio and electrical conductivity of the alloy subjected during homogenization. UTS: ultimate tensile strength; YS: yield strength.

| Homogenization Time (h) | UTS (MPa) | YS (MPa) | Yield Ratio (YS/UTS) | Conductivity (MS/m) |
|---|---|---|---|---|
| 0 | 212.5 | 172.1 | 0.81 | 6.52 |
| 8 | 143.5 | 101.9 | 0.71 | 7.31 |
| 12 | 137.4 | 93.4 | 0.68 | 7.78 |
| 16 | 133.5 | 89.4 | 0.67 | 8.05 |
| 20 | 129.6 | 81.6 | 0.63 | 8.27 |
| 24 | 124.8 | 76.1 | 0.61 | 8.48 |
| 28 | 119.6 | 67.0 | 0.56 | 8.56 |
| 32 | 116.5 | 60.6 | 0.52 | 8.66 |

The homogenization treatment promoted the precipitation of $T$ phases and $Al_3(Zr_xTi_yLi_{1-x-y})$ dispersoid particles and caused the diffusion of Cu, Mn, Zr, Li and Ti atoms in the Al matrix. The effect of solute atoms on the resistivity is usually well described by Matthiessen's law, which revealed that the conductivity is related to the concentration of the solute atoms. Moreover, Mg and Zn atoms also diffused into the lattices of $T$ phases and $Al_3(Zr_xTi_yLi_{1-x-y})$ dispersoid particles during the high temperature homogenization, which further reduced the solid solubility of the Al matrix and therefore led to the increase of electrical conductivity. Besides, although the solubility of Zr in AlLi is negligible, the $Al_3Zr$ phase can take in up to 1.3 at.% Li [35], that is to say, alloying elements such as Li and Mg partitioned between the matrix and some of the dispersoid particles, for example, $Al_3(Zr_xTi_yLi_{1-x-y})$, resulting in improved ductility. Meanwhile, the precipitation of the coherent $Al_3(Zr_xTi_yLi_{1-x-y})$ phases can reduce the misfit degree of coherent interface and make the strength distribution of bonding interface more uniform, which was beneficial to improve the heat stability. $T$ phase is also a kind of toughening phase, which can effectively improve the workability of the alloy. At the same time, the dissolution of coarse constituent particles leaded to weakening of the grain boundary strengthening effect, so the mechanical strength of the alloy decreased.

Figure 12 showed the DSC curves of as-cast and homogenized alloys. It can be seen that the as-cast alloys had two endothermic peaks at 521.9 °C and 649.9 °C, respectively. The endothermic peak at 521.9 °C was largely caused by the dissolution reaction of Li-containing low-melting eutectic phases, which were widely distributed in the dendrite and dendrite gaps. The second endothermic peak was due to the melting alloy, which means the melting point of the as-cast alloy might achieve to 649.9 °C. After second-step homogenization, a decrease in the volume fraction of Li-containing low-melting eutectic phases leads to a significant decrease in the intensity of the first endothermic peak up to a tiny value, which cannot be ignored due to the possibility of endothermic reaction of the remained AlCuFeMn phase. It is worth noting that there was a second tiny endothermic peak at 544.2 °C in the homogenized alloy, which was presumed to be the dissolution reaction of $Al_3(Zr_xTi_yLi_{1-x-y})$ dispersoid particles because of the strong Al-Zr covalent bond. Compared to the as-cast alloy, the melting endothermic peak of the homogenized alloy shifts to the right, where the peak value rose from 649.9 °C to 654.5 °C. The $Al_3(Zr_xTi_yLi_{1-x-y})$ and $T$ phases played a significant role as pinning the dislocations and sub-grain boundaries near the dendrites and grain boundaries, which subsequently preventing grain boundary moving and grain merging [36]. Therefore, they had a strong influence on retarding recrystallization and grain growth and consequently on grain size control and improvement on thermal stability as well as the dissolution reaction of the low-melting eutectic phases, which finally resulted in the significant improvement of the thermal stability.

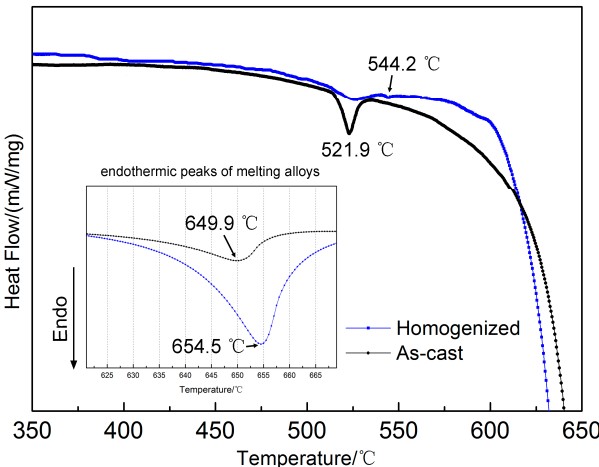

**Figure 12.** DSC curves of as-cast and homogenized alloys.

## 5. Conclusions

The microstructural evolution and properties of an Al-Cu-Li alloy with Mg and Zn microalloying ingot subjected to homogenization were investigated. The following conclusions can be drawn:

(1) Low-melting eutectic phases such as $S$, $T_2$, $T_B$ and coarse $T_1$ phases were nucleated in and around the dendrites of the as-cast alloy, in which the early enrichment of Mg-Zn atoms clusters was revealed.

(2) The Li-containing phases dissolves preferentially during homogenization followed by the diffusion of Mg, Zn, Zr, and Mn, leading to the dissolution of $S$ and AlCuFeMn phases and the precipitation of $T$ phases as well as $Al_3Zr$ particles.

(3) Mg-Zn atom clusters was easily bound to the vacancies together thus promoted the nucleation $Al_3(Zr_xTi_yLi_{1-x-y})$ dispersoid particles, which resulted in the non-recrystallization characteristic of the alloy.

(4) After homogenization, the yield ratio was decreased from 0.81 to 0.52, displaying better plastic deformation ability, in which the formation of $Al_3(Zr_xTi_yLi_{1-x-y})$ dispersoids resulted in the significant improvement on thermal stability of the alloy.

**Author Contributions:** Data Curation, X.W.; Formal Analysis, W.Y.; Investigation, W.Y. and X.W.; Methodology, R.D. and W.Y.; Project Administration, H.L.; Supervision, H.L.; Writing-Original Draft, W.Y.; Writing-Review & Editing, H.L. and R.D.

**Funding:** This research received no external funding.

**Acknowledgments:** Scientific/technical assistance from Institure for Materials Microstructure at CSU is acknowledged.

**Conflicts of Interest:** The authors declare no conflict of interest.

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
