# Peer review of "Investigation on Microstructural Evolution and Properties of an Al-Cu-Li Alloy with Mg and Zn Microalloying during Homogenization"

_metals, doi:10.3390/met8121010_

Reviewer 1 Report

Font size in most figures are not appropriate and thus the text in figures are not readable, including some scale bars.

EDS measurements on non-homogeneous microstructures always contain errors. Besides, depending on the Acc. Vol., the EDS information comes from an interaction volume under the surface. Additionally, depending on the beam size, there would be some informaiton from the surrounding area. So you see that point analysis in EDS contains a lot of errors, which has not been discussed in the paper. Specifically in Table 2, I would like to see Standard deviation for each EDS measurement and each element. What is the "resolution" of EDS for elemental quantificaiton? Fe content in some alloys is in the range of 0.1-0.2%. Is that really a reliable EDS measurement? ... Therefore, the whole dicussion around Table 2 and the type of particles, in the current format, are questionable!

What is the meaning of "Whole time" in table 3?

Author Response

Response to Reviewer 1 Comments

Thank you for the reviewer comments concerning our manuscript entitled “Investigation on microstructural evolution and properties of an Al-Cu-Li alloy with Mg and Zn microalloying during homogenization” (manuscript ID: metals-387124). Those comments are all valuable and helpful for revising and improving our paper. We have studied comments carefully and have made correction which we hope meet with approval. Revised portions are highlighted in yellow in the revised manuscript. The responds to the reviewer’s comments as well as main corrections in the paper are as follows. In addition, the lines of the revised statements in the revised manuscript are shown in brackets.

Point 1: Font size in most figures are not appropriate and thus the text in figures are not readable, including some scale bars.

Response 1: Thanks for your comments. According to your comments, we changed the size of fonts in some explanatory text and scale bars. Also, the location of some pictures was changed for the readability of the text in figures. The revised pictures are as follows:

Section 3.2: (line 148-151)

Figure 4. Elements mapping of the selected regions in as-cast, first-step homogenized (400, 8h) and second-step homogenized (400, 8h+500, 24h) alloys by EPMA: (a)(b)(c) Cu, (d)(e)(f) Mg and (g)(h)(i) Zn.

Section 3.2: (line 173-175)

Figure 5. Elements mapping of the as-cast alloy by EPMA: (a) Cu, (b) Mg, (c) Zn, (d) Zr, (e) Mn and (f) Fe.

Section 3.3: (line 197-200)

Figure 6. SEM images on microstructures of the alloys (a) as-cast and homogenized at (b) 400 for 8h subsequent to second-step homogenization at 500 for (c) 4h, (d) 8h, (e) 12h, (f) 16h and (g) 24h.

Section 3.4: (line 241-244)

Figure 7. TEM images on Al3Zr phases with (a) diffraction pattern with zone axis of [110]Al through (b) bright field and (c) dark field, (d) STEM-HAADF and elements mapping of (e) Al and (f) Zr in the particle and (g) HRTEM image of the fine precipitates with FFT.

Section 3.4: (line 256-258):

Figure 8. TEM images of (a) phases in the homogenized alloy and EDS analysis of (b) point A, (c) point B and (d) point C in the rod-like phases.

Point 2: EDS measurements on non-homogeneous microstructures always contain errors. Besides, depending on the Acc. Vol., the EDS information comes from an interaction volume under the surface. Additionally, depending on the beam size, there would be some informaiton from the surrounding area. So you see that point analysis in EDS contains a lot of errors, which has not been discussed in the paper. Specifically in Table 2, I would like to see Standard deviation for each EDS measurement and each element. What is the "resolution" of EDS for elemental quantificaiton? Fe content in some alloys is in the range of 0.1-0.2%. Is that really a reliable EDS measurement? ... Therefore, the whole dicussion around Table 2 and the type of particles, in the current format, are questionable!

Response 2: Thank you for your kind comments. As we all know, EDS measurement is a qualitative and semi-quantitative analysis method, containing a lot of errors. This point was fully considered and discussed when we designed the experiments. Unfortunately, we did not mention it in the manuscript, which is our mistake. Actually, in order to reduce the measurement errors of EDS, we chose an area test instead of a single point test. So, it is not suitable for the point A-K in the original manuscript, which has been changed to the area A-K.  Besides, energy dispersive X-ray spectroscopy GENESIS 60S with the resolution of 131eV was used. We obtained a precision around ± 1% owing to factors such as uncertainties in the compositions of the standards and errors in the various corrections which need to be applied to the raw data. So, we revised the data in table 2 and provided only one decimal place in all EDS data now. In fact, we have detected Fe content in area A-K. But in the area E and J, the Fe content detected was nearly 10 times that of other areas with a significant increase of Mn content, which led to the judgement that the closet phase may be the insoluble AlCuFeMn phase. When detected these phases, phase diagrams, appearance, size and contrast of the phase as well as the previous work were all considered. EDS is only one of the experimental evidences. Suitable references have been added to support our judgement in the revised manuscript. After all, the corrections we made in revised manuscript are as follows:

Section 3.3: (line 197-200)

Figure 6.SEM images on microstructures of the alloys (a) as-cast and homogenized at (b) 410℃ for 8h subsequent to second-step homogenization at 500℃ for (c) 4h, (d) 8h, (e) 12h, (f) 16h and (g) 24h.

Section 2: (line 94-98)

"Phase constitution and evolution during homogenization were detected by scanning electron microscope (SEM) Quanta-200 with Energy dispersive X-ray Spectroscopy (EDS) GENESIS 60S with the resolution of 131ev. A precision around ± 1% was obtained owing to factors such as uncertainties in the compositions of the standards and errors in the various corrections which need to be applied to the raw data. "

Section 3.3 (line 201):

Table 3 Chemical composition of areas in Figure 6 (at.%).

Area

Al

Cu

Mg

Zn

Mn

Fe

Zr

Closet phase

A

67.6

29.3

1.5

1.2

0.3

0.2

-

Al2Cu

B

70.1

15.1

13.6

0.9

0.2

0.2

-

Al2CuMg

C

61.7

34.6

1.4

1.3

0.4

0.6

-

Al7.5Cu4Li

D

80.1

13.4

5.1

0.5

0.1

0.2

-

Al6CuLi3

E

79.7

11.3

1.6

0.4

2.4

4.6

-

AlCuFeMn

F

61.4

30.7

3.3

3.2

0.5

0.7

-

Al2CuLi

G

64.8

31.8

1.4

1.0

0.1

0.2

~0.5

Al2Cu

H

64.0

32.0

0.9

1.2

0.3

0.4

~0.5

Al2Cu

I

65.9

30.3

1.9

1.0

0.1

0.2

-

Al2Cu

J

75.2

11.9

0.3

0.2

5.5

6.7

-

AlCuFeMn

K

67.1

29.6

1.3

0.8

0.1

0.3

~0.5

Al2Cu

Section 3.3: (line 184-192)

"As shown in point A, the atomic ratio of Al and Cu atoms in the long and coarse phase is close to 2:1, which was formed almost on the dendrites and grain boundaries. The closest phase might be determined to be θ phase (Al2Cu) [22]. The atomic ratio of Cu and Mg atoms in the gray eutectic phase of point B is close to 1:1, therefore the closest phase might be S phase (Al2CuMg) [23]. In the bright phase with a larger contrast of point C, EDS analysis revealed a high content of Cu, where the atomic ratio of Al and Cu atoms is close to 7:4. Because lithium is too light to be detected, the phase of point C might be identified as TB phase (Al7.5Cu4Li) [16]. In the white particle of point D on the dendrite, the atomic ratio of Al to Cu atoms is close to 6:1, which might be T2 (Al6CuLi3) [24]. "

Section 3.3: (line 194-196)

"In area F, the atomic ratio of Al and Cu atoms in the rod-like phases is close to 2:1, which is closest to T1 (Al2CuLi) phase judged by the appearance and element composition synthetically [22]."

References: (line 430-434)

22.  Yoshimura, R.; Konno, T.; Abe, E.; Hiraga, K. Transmission electron microscopy study of the evolution of precipitates in aged Al–Li–Cu alloys: the θ′ and T1 phases, Acta. Mater. 2003, 51, 4251–4266.

23.  Wang, S.; Starink, M. Two types of S phase precipitates in Al–Cu–Mg alloys, Acta. Mater. 2007, 55, 933–941.

24.   Li, J.; Li, C.; Peng, Z.; Chen, W.; Zheng, Z. Corrosion mechanism associated with T1 and T2 precipitates of Al–Cu–Li alloys in NaCl solution, J. Alloy. Compd. 2008, 460, 688-693.

Point 3: What is the meaning of "Whole time" in table 3?

Response 3: The "Whole time" in table 3 means the whole homogenization time. For example, the whole time of 8h means homogenization at 400 for 8h. The whole time of 12h means homogenization at 400 for 8h following by homogenization at 500 for 4h and so on. We should have explained it clearly because this was the first time we used "Whole time". To avoid misunderstanding, we revised the text in table 3 and changed the "Whole time" to the "homogenization time" in order to correspond to the x-coordinate in figure 10. The corresponding changes made in the revised manuscript are as follows:

Section 4.3: (line 317,318)

"The homogenization time of 12h means homogenization at 400 for 8h following by homogenization at 500 for 4h and so on. "

Section 4.3(line 341, 342):

Table 4 Mechanical properties, yield ratio and electrical conductivity of the alloy subjected during homogenization.

Homogenization   time (h)

UTS (MPa)

YS (MPa)

Yield ratio (YS/UTS)

Conductivity (MS/m)

0

212.5

172.1

0.81

6.52

8

143.5

101.9

0.71

7.31

12

137.4

93.4

0.68

7.78

16

133.5

89.4

0.67

8.05

20

129.6

81.6

0.63

8.27

24

124.8

76.1

0.61

8.48

28

119.6

67.0

0.56

8.56

32

116.5

60.6

0.52

8.66

We appreciate for Editors/Reviewers’ warm work earnestly, and hope that the correction will meet with approval. Once again, thank you very much for your comments and suggestions.

Corresponding author: Hongying Li

Reviewer 2 Report

The subject of the article is of high importance. The aim of the study is well described and the experiments are well designed and performed. Unfortunately, in parts the article lacks from language difficulties (use of tense, singular/plural, special terms). This makes it difficult to understand the discussion and conclusion sections.

Some special issues are addressed in the following.

Section 2 Materials and Methods

In the section and in the corresponding diagram it does not become clear when the quench steps were performed. The sample preparation should always be described before the description of the corresponding methods and devices for the investigation. Mechanical testing should be described in more detail (lines 104/105).

Line 93 corroded? etched?

Section 3.1.:

“dendritic grain” is not an appropriate term

Section 3.2:

Please refer to the color legend of the maps and not to “lighter colors”

Conclusion in lines 148/149 is not clear – possibly due to language problems

Line 153: you describe the elemental composition of grain boundary precipitations, not of the grain boundary itself. Areas A, B and C are marked in Figure 4a (not in 4b). Concerning your quantitative EDS results: for major elements you can obtain a precision around ± 1% (owing to factors such as uncertainties in the compositions of the standards and errors in the various corrections which need to be applied to the raw data). Therefore I highly recommend providing only one decimal place in all EDS data. Especially the Zr concentration should be given as ~0.5 at%. Accordingly, I recommend to give all the concentrations in the text in at% as done in the table 2 (where rounding off one decimal place is also recommended).

Sections 3.3. Intermetallic constituent particles / 3.4. Dispersoid particles

Please note: All precipitations are intermetallic compounds, some are constituent particles and some are dispersoids. This also links these sections to section 3.2. and should be done by using the terms.

The resolution in Figure 5 is too low – the areas marked by capital letters are not visible.

Results in Figure 6 are very impressive, but not well shown. Show Fig6 a and b the same area and magnification? What does the box in Fig.6b mean? Can you move the diffraction pattern in fig.6 a to the left?

Section 4. Discussion

In section 4.1. you introduce new results obtained by XRD. This should be another results section. In doing so this section should be merged with the section 4.2. Discussion of microstructure evolution.

Section 4.3. also introduces additional results (electrical and mechanical properties). To be consistent with the article’s organization the section also should be divided into results and discussion.

Author Response

Response to Reviewer 2 Comments

Thank you for the reviewer comments concerning our manuscript entitled “Investigation on microstructural evolution and properties of an Al-Cu-Li alloy with Mg and Zn microalloying during homogenization” (manuscript ID: metals-387124). Those comments are all valuable and helpful for revising and improving our paper. We have studied comments carefully and have made correction which we hope meet with approval. Revised portions are highlighted in yellow in the revised manuscript. The responds to the reviewer’s comments as well as main corrections in the paper are as follows. In addition, the lines of the revised statements in the revised manuscript are shown in brackets.

Point 1: Section 2 Materials and Methods

In the section and in the corresponding diagram it does not become clear when the quench steps were performed. The sample preparation should always be described before the description of the corresponding methods and devices for the investigation. Mechanical testing should be described in more detail (lines 104/105).

Response 1: Thank you for your comments. We think that the quench steps are very important because it will affect final microstructures. All of our results are based on the experiments with water quench step after homogenization. We apologize for not mentioning the details of sample preparation. Besides, more details of mechanical properties including sample size have been added to the revised manuscript. Those corrections are as follows:

Section 2: (line 82-88)

"The rectangular ingot (220mm×140mm×35mm) from the factory was detected for the present researches, whose chemistry was shown in Table 1. The cube samples (1.5mm×1.5mm×1.5mm) cut from the central part of the ingot were subjected to a two-step homogenization in salt bath furnace, in which the samples were heated at 400 for 8h followed by 500 for 24h and cooled by water quench. The schematic of the thermal history for the homogenization was shown in Figure 1, in which highlights (with dots) were the times of samples taken from the furnace to determine the evolution of microstructure. "

Section 2: (line 107-110)

"The mechanical properties were measuring by a universal testing machine MTS 858 with a strain rate of 1.0×10-3 s-1. The samples (figure 2) were cut from the ingots and homogenized under various conditions before the tensile test. Both results took the average of 5 samples. "

Section 2: (line 111-112)

Figure 2 Tensile test sample of the alloy.

Point 2: Line 93 corroded? etched?

Response 2: There is a mistake in this sentence. What we want to describe is just electrolytic polishing. The revised text is as follow: "The samples were subjected to electrolytic polishing by HBF3 (16.8g/L). " (line 93-94)

Point 3: “dendritic grain” is not an appropriate term

Response 3: Thank you for your comment. According to your comment, we will pay more attention to the usage of special terms in the future. The "dendritic grain" has been revised as "dendrites". (line 116)

Point 4: Section 3.2:

Please refer to the color legend of the maps and not to “lighter colors”

Response 4: Thank you for your comment. We found that readers may feel confused about the statements in section 3.2. We have not displayed our results well due to the poor English manner. "The text in Section 3.2 has been revised and the "lighter color" has been deleted as follow: The elements distribution of Cu, Mg and Zn was investigated by EPMA in the as-cast, first-step homogenized (400, 8h) and two-step homogenized (400, 8h+500, 24h) alloys in Figure 4, where the concentration of alloying elements in different regions was listed next to the Figure. " (line 126-128)

Point 5: Conclusion in lines 148/149 is not clear – possibly due to language problems

Response 5: Thank you for your comment. Actually, we wanted to illustrate the difference between Cu, Mg and Zn on enrichment phenomenon. This conclusion has been revised as "The results revealed that Cu and Mg atoms tended to move from the coarse phases on the dendrites towards the Al matrix during homogenization treatment, while the movement of Zn atoms was opposite, from the Al matrix towards the second phases." (line 144-147)

Point 6: Line 153: you describe the elemental composition of grain boundary precipitations, not of the grain boundary itself. Areas A, B and C are marked in Figure 4a (not in 4b). Concerning your quantitative EDS results: for major elements you can obtain a precision around ± 1% (owing to factors such as uncertainties in the compositions of the standards and errors in the various corrections which need to be applied to the raw data). Therefore I highly recommend providing only one decimal place in all EDS data. Especially the Zr concentration should be given as ~0.5 at%. Accordingly, I recommend to give all the concentrations in the text in at% as done in the table 2 (where rounding off one decimal place is also recommended).

Response 6: Thank you for your comments. EDS measurements contained a lot of errors, which is not mentioned in the manuscript before. According to your comments, we have corrected the table 2 and provided the concentrations of area A-C in figure 4 in another table. Those corrections are as follows:

Section 3.3: (line 201)

Table 3 Chemical composition of areas in Figure 6 (at.%).

Area

Al

Cu

Mg

Zn

Mn

Fe

Zr

Closet   phase

A

67.6

29.3

1.5

1.2

0.3

0.2

-

Al2Cu

B

70.1

15.1

13.6

0.9

0.2

0.2

-

Al2CuMg

C

61.7

34.6

1.4

1.3

0.4

0.6

-

Al7.5Cu4Li

D

80.1

13.4

5.1

0.5

0.1

0.2

-

Al6CuLi3

E

79.7

11.3

1.6

0.4

2.4

4.6

-

AlCuFeMn

F

61.4

30.7

3.3

3.2

0.5

0.7

-

Al2CuLi

G

64.8

31.8

1.4

1.0

0.1

0.2

~0.5

Al2Cu

H

64.0

32.0

0.9

1.2

0.3

0.4

~0.5

Al2Cu

I

65.9

30.3

1.9

1.0

0.1

0.2

-

Al2Cu

J

75.2

11.9

0.3

0.2

5.5

6.7

-

AlCuFeMn

K

67.1

29.6

1.3

0.8

0.1

0.3

~0.5

Al2Cu

Section 3.2: (line 176)

Table 2 Element concentration in areas A-C in Figure 5 (%).

Area

Cu

Mg

Zn

Mn

Zr

Fe

A

45.6

0.5

2.2

-

-

-

B

27.5

0.4

1.2

-

~0.5

-

C

20.9

-

0.7

7.5

-

4.6

Point 7: Sections 3.3. Intermetallic constituent particles / 3.4. Dispersoid particles

Please note: All precipitations are intermetallic compounds, some are constituent particles and some are dispersoids. This also links these sections to section 3.2. and should be done by using the terms.

Response 7: Thank you for you comments. The sub-titles are so important and we have revised the sub-titles to link the sections together according to your comments. Here are the corrections:

Section 3.3 (line 177):

"3.3 Constituent particles"

Section 3.4 (line 219):

"3.4 Dispersoids"

Point 8: The resolution in Figure 5 is too low – the areas marked by capital letters are not visible.

Response 8: Thank you for your comment. We feel so sorry for the low resolution in Figure 5. Something was wrong when we uploaded the manuscript. We also changed the color and font size of text in figures. The corrected figures are as follows: (line 197-200)

Figure 6. SEM images on microstructures of the alloys (a) as-cast and homogenized at (b) 400 for 8h subsequent to second-step homogenization at 500 for (c) 4h, (d) 8h, (e) 12h, (f) 16h and (g) 24h.

Point 9: Results in Figure 6 are very impressive, but not well shown. Show Fig6 a and b the same area and magnification? What does the box in Fig.6b mean? Can you move the diffraction pattern in fig.6 a to the left?

Response 9: Thank you for your comments. The TEM figures are so important for the paper, which are the experimental evidence of Al3Zr particles. Figure 6a shows the bright field of Al3Zr particles. Figure 6b is the same area and magnification as figure 6a. Some tiny bright spots were observed except Al3Zr particles in figure 6b. The square was used for readers to be able to find the tiny precipitates since they are just like stars in the Milky Way. According to your comment, we have set the diffraction pattern as the new figure 6a to the left. The revised parts are as follows:

Section 3.4: (line 225,226)

"According to the dark field image in Figure 7c, in addition to the large spherical particles, a large number of tiny bright spots were also observed in the yellow square. "

Section 3.4: (line 241-244)

Figure 7. TEM images on disperoids with (a) diffraction pattern with zone axis of [110]Al through (b) bright field and (c) dark field, (d) STEM-HAADF and elements mapping of (e) Al and (f) Zr in the particle and (g) HRTEM image of the fine precipitates with FFT.

Point 10: Section 4. Discussion

In section 4.1. you introduce new results obtained by XRD. This should be another results section. In doing so this section should be merged with the section 4.2. Discussion of microstructure evolution.

Response 10: Thank you for your comments. According to your comments, the section 4.1 has been merged with the section 4.2. And the new sub-title of the new section has been revised as "4.1 Microstructural evolution analysis" (line 262). The results of XRD have been introduced first following by the schematic illustration as well as the discussion of microstructural evolution. We find that such an organization is more reasonable and logical. The revised paragraphs are as follows:

Section 4.1: (line 260-313)

4.1 Microstructural evolution analysis

Dendritic grains were observed on the as-cast alloy as shown in Figure 3, which should be the result of composition fluctuations during the non-equilibrium solidification of the alloy [24]. Some intermetallic constituent particles were found formed during solidification of the molten alloy at relatively high temperatures, and their sizes lie in the range of one to several tens of microns. The limited solubility of Li in Al is 4wt.%, which decreases to 1wt.% at room temperature. Due to the lightness, Li was difficult to detect by EPMA and EDS, so Li was not found in the elemental analysis of as-cast alloys. However, the Li content in as-cast alloy is 1.3%-1.8%, so the supersaturation of Li in the ingot after solidification was very high, which means the easy formation of coarse primary phases. Those large number of rod-like phases around the grain boundaries were presumed as primary phases of T1 as shown in Figure 4 and 6. Furthermore, XRD analysis of the as-cast alloy showed that TB and T2 were also existed in Figure 9. This is because the chemical property of Li is very active, which resulted in nucleation intermetallic constituent particles with Cu on the grain boundaries during the solidification process. XRD also showed the presence of Al6Mn in the as-cast alloy. This primary phase is often accompanied by the presence of Al6(MnFe), which was also not observed in the SEM. It is worth noting that these Li-containing primary phases (T1, TB and T2) were also enriched with clusters of Mg-Zn atoms.

Figure 9. XRD analysis of as-cast alloys and alloys subjected to different homogenization.

The limited solubility of Cu in the Al matrix is 5.67 wt.%, which can improve the cutting performance of the alloy as well as certain solid solution and precipitation strengthening. Figure 4 showed that there was a large amount of Cu on the grain boundaries in the as-cast microstructure, forming dendritic segregation. This is because the process of uniform diffusion of Cu atoms in the solid phase occurred slowly, which easily formed coarse intermetallic constituent particles like Al2Cu on the grain boundaries together with Al atoms [25]. When Mg and Cu were simultaneously added, S (Al2CuMg) intermetallic constituent phases were easily formed and detected in Figure 6 [26,27]. Mg and Zn were detected to concentrate on almost all over the precipitates and dendrites because Mg and Zn atom clusters were formed during solidification of the as-cast alloy, which adsorbed a large number of free atoms in matrix and promoted the formation of primary phases. Therefore, the dendrites in the as-cast alloys are mainly caused by the non-uniform distribution of components formed by Cu, Mg and Zn. The addition of Mn and Zr will inhibit recrystallization obviously and improve the workability and corrosion resistance of Al-Li alloy [28]. Coarse primary phases with Zr were formed during solidification, meanwhile, Mn was generally detected in low-melting eutectic phases at the dendritic gaps with the impurity Fe [29]. Those coarse insoluble constituent particles have minor effects on strength but adversely affect ductility and fracture toughness, which were easily melted in the subsequent hot deformation process, forming crack initiation sites and resulting in rolling cracks [30].

Based on the investigation results above, the schematic illustration on microstructural evolution during homogenization was displayed on Figure 10. The intermetallic constituent particles form during solidification of the molten alloy at relatively high temperatures, and their sizes lie in the range of one to several tens of microns such as Al2Cu, Al2CuMg and AlCuFeMn phases. Many kinds of Li-containing eutectic phases with Mg-Zn atom clusters were nucleated in the dendrites, which was surrounded by primary coarse T1 phases. After first-step homogenization, T2, TB, and T1 phases dissolve preferentially. However, only a small amount of Al2Cu phase dissolved due to the low diffusion rate of Cu. Mg atoms diffused into the matrix from the primary phases and Zn diffused to the Cu-containing phases, accompanied by the combination between Mg-Zn atom clusters and vacancies. After second-step homogenization for 12h, only Fe-containing AlCuFeMn particles still existed. The T phases and Al3(ZrxTiyLi1-x-y) dispersoid particles nucleated at the Mg-Zn atom clusters. Finally, dendritic segregation was completely eliminated by step homogenization. Already formed T phase and Al3(ZrxTiyLi1-x-y) particles grew further and fine and uniform Al3(ZrxTiyLi1-x-y) particles were newly generated. The alloy showed non-recrystallization characteristic and improvement on hot workability after homogenization.

Figure 10. Schematic illustration on microstructures of the alloy during homogenization

Point 11: Section 4.3. also introduces additional results (electrical and mechanical properties). To be consistent with the article’s organization the section also should be divided into results and discussion.

Response 11: Thank you for your comments. According to your comment, section 4.3 has been divided into results and discussion. The title of section 4.3 has also been changed as "4.2 properties evolution analysis" (line 316) to be consistent with the organization. The revised paragraphs are as follows:

Section 4.2: (314-362)

4.2 Properties evolution analysis

The mechanical properties and electrical conductivity of the alloy during homogenization were detected in Figure 11, and the value of ultimate tensile strength (UTS), yield strength (YS), yield ratio and electrical conductivity were list in Table 4. The homogenization time of 12h means homogenization at 400 for 8h following by homogenization at 500 for 4h and so on. As the homogenization proceeds, the UTS of the alloy decreased sharply from 212.5MPa to 143.5MPa, while the YS decreased from 172.1MPa to 101.9MPa at 400 for 8h as the yield ratio decreased from 0.81 to 0.71. Prolonging homogenization time at 500, the alloy continued to soften as the yield ratio decreased to 0.52. As the homogenization proceeds, the electrical conductivity of the alloy continued to increase from 6.52 MS/m to 8.66 MS/m.

The homogenization treatment promoted the precipitation of T phases and Al3(ZrxTiyLi1-x-y) dispersoid particles and caused the diffusion of Cu, Mn, Zr, Li and Ti atoms in the Al matrix. The effect of solute atoms on the resistivity is usually well described by Matthiessen’s law, which revealed that the conductivity is related to the concentration of the solute atoms. Moreover, Mg and Zn atoms also diffused into the lattices of T phases and Al3(ZrxTiyLi1-x-y) dispersoid particles during the high temperature homogenization, which further reduced the solid solubility of the Al matrix and therefore led to the increase of electrical conductivity. Besides, although the solubility of Zr in AlLi is negligible, the Al3Zr phase can take in up to 1.3 at. % Li [31], that is to say, alloying elements such as Li and Mg partitioned between the matrix and some of the dispersoid particles, for example, Al3(ZrxTiyLi1-x-y), resulting in improved ductility. Meanwhile, the precipitation of the coherent Al3(ZrxTiyLi1-x-y) phases can reduce the misfit degree of coherent interface and make the strength distribution of bonding interface more uniform, which was beneficial to improve the heat stability. T phase is also a kind of toughening phase, which can effectively improve the workability of the alloy. At the same time, the dissolution of coarse constituent particles leaded to weakening of the grain boundary strengthening effect, so the mechanical strength of the alloy decreased.

Figure 11. Mechanical properties and electrical conductivities of the alloy during homogenization

Table 4 Mechanical properties, yield ratio and electrical conductivity of the alloy subjected during homogenization.

Homogenization time (h)

UTS (MPa)

YS (MPa)

Yield ratio (YS/UTS)

Conductivity (MS/m)

0

212.5

172.1

0.81

6.52

8

143.5

101.9

0.71

7.31

12

137.4

93.4

0.68

7.78

16

133.5

89.4

0.67

8.05

20

129.6

81.6

0.63

8.27

24

124.8

76.1

0.61

8.48

28

119.6

67.0

0.56

8.56

32

116.5

60.6

0.52

8.66

Figure 12 showed the DSC curves of as-cast and homogenized alloys. It can be seen that the as-cast alloys had two endothermic peaks at 521.9 and 649.9, respectively. The endothermic peak at 521.9 was largely caused by the dissolution reaction of Li-containing low-melting eutectic phases, which were widely distributed in the dendrite and dendrite gaps. The second endothermic peak was due to the melting alloy, which means the melting point of the as-cast alloy might achieve to 649.9. After second-step homogenization, a decrease in the volume fraction of Li-containing low-melting eutectic phases leads to a significant decrease in the intensity of the first endothermic peak up to a tiny value, which cannot be ignored due to the possibility of endothermic reaction of the remained AlCuFeMn phase. It is worth noting that there was a second tiny endothermic peak at 544.2°C in the homogenized alloy, which was presumed to be the dissolution reaction of Al3(ZrxTiyLi1-x-y) dispersoid particles because of the strong Al-Zr covalent bond. Compared to the as-cast alloy, the melting endothermic peak of the homogenized alloy shifts to the right, where the peak value rose from 649.9°C to 654.5°C. The Al3(ZrxTiyLi1-x-y) and T phases played a significant role as pinning the dislocations and sub-grain boundaries near the dendrites and grain boundaries, which subsequently preventing grain boundary moving and grain merging [32]. Therefore, they had a strong influence on retarding recrystallization and grain growth and consequently on grain size control and improvement on thermal stability as well as the dissolution reaction of the low-melting eutectic phases, which finally resulted in the significant improvement of the thermal stability.

Figure 12. DSC curves of as-cast and homogenized alloys.

We appreciate for Editors/Reviewers’ warm work earnestly, and hope that the correction will meet with approval. Once again, thank you very much for your comments and suggestions.

Corresponding author: Hongying Li

Reviewer 3 Report

1. "Three different distributions of Cu concentration were observed obviously on the grain boundary in Figure 4b..."  This and many references to figures in the same paragraph are wrong. Please correct. 

2. Are the red arrows in Fig. 4e pointing at a precipitate free zone or a solute free zone? Since the mapping shows a lack of Mn signal, I would say it is a solute free zone.

3. Section 3.2 (segregation evolution) is written in a very confusing manner. The numbers next to the micrographs are not legible.

4. Section 3.3 (intermetallic constituent particles) is mislabeled as 3.2. 

5. The labels on Figure 5 are not legible. Please use a different color and larger font size. It is nearly impossible to locate A-K, except for G and H.

6. In Section 3.4, the authors talk about crystallographic directions and planes. Do not use comma between indices, so it should be [110] and [1,1,0]. Also negative numbers are represented with a bar above that number so do not use -1. 

7. "The Al3Zr primary phase does not meet this characteristic due to its short number and large size." What does short number mean? Also, are you referring to 20-30 nm precipitates as large particles? Primary Al3Zr precipitates, formed during solidification, can be a few microns across and visible with an optical microscope. Another issue is the fact that this is a homogenized alloy with the highest homogenization temperature being 500 C; it is completely possible that these precipitates could have formed during these heat treatments. There are studies aging Zr-containing Al alloys at temperatures as high as 450 C. 

8. Use of English must be improved. Please work with a professional language editor.

Author Response

Response to Reviewer 3 Comments

Thank you for the reviewer comments concerning our manuscript entitled “microstructural evolution and properties of an Al-Cu-Li alloy with Mg and Zn microalloying during homogenization 

"Three different distributions of Cu concentration were observed obviously on the grain boundary in Figure 4b..."  This and many references to figures in the same paragraph are wrong. Please correct. 

Thank you for your comments. We are so sorry to make those stupid mistakes. We have checked and reviewed the manuscript more carefully to avoid those mistakes. The revised statements are as follows:

Section 3.2: (line 154-172)

"Three different distributions of Cu concentration were observed obviously on the grain boundary in Figure 5a, which were divided into area A, B and C. The element concentration in areas A-C can be obtained in Table 2. The average concentration of Cu atoms was the highest (45.6%) at area A, where the concentrations of Mg and Zn were also relatively highest. No enrichment of Zr, Mn and Fe were observed at area A. The average concentration of Cu atoms at point B was 27.5%, where the enrichment of Mg and Zn were apparent with lower concentration than that in area A. The enrichment of Mn and Fe was found in area B, where the color was obviously brighter than the surrounding in Figure 5e and f. It is worth noting that there was enrichment of Zr in bright color with the average concentration of ~0.5% at area B, which was pointed by yellow arrow in Figure 5d. Besides, the color of Fe in the particle pointed by the arrow was also different from the surrounding in Figure 5f, indicating that the concentration of Fe was higher here. The average concentration of Cu atoms in area C was 20.9%, where Mg was hardly present because the color of Mg was the darkest in Figure 5b. The existence of Zn atoms was observed but the enrichment was not formed because the color of Zn at area C was very uniform with the grain. The Mn and Fe atoms were enriched in area C and have performed the highest average concentration relatively. Moreover, unlike Cu, Mg, Zn, Zr and Fe, the distribution of Mn atoms in the grain was not uniform in Figure 5e. According to previous work, the toughness phase, T (Al20Cu2Mn3) phase [21], might formed in grains and promoted the solute free zone along the grain boundary in darker color shown in Figure 5e."

Point 2:  

Thank you for your comments. The element distribution can be well characterized by EPMA. The mapping absolutely shows a lack of Mn atoms, not the precipitates. According to your comments, the "precipitate free zone" is not suitable here. We have revised the statement and figures as follow: "Moreover, unlike Cu, Mg, Zn, Zr and Fe, the distribution of Mn atoms in the grain was not uniform in Figure 5e. According to previous work, the toughness phase, T (Al20Cu2Mn3) phase [21], might formed in grains and promoted the solute free zone along the grain boundary in darker color shown in Figure 5e." (line 169-175)

Figure 5. Point 3:  

Thank you for your comments. We feel so sorry to make the reviewer confused due to the poor English manner. The section 3.2 has been revised to show the results of elements distribution of Cu, Mg and Zn by EPMA more clearly. The font size of text in figure 3 is also been changed to be readable. The revised  

"The elements distribution of Cu, Mg and Zn was investigated by EPMA in the as-cast, first-step homogenized (400, 8h+500. The segregation of Mg around the dendrites still existed in Figure 4e. But many bright particles represented for phases with Mg enrichment were visible in the grains, indicating that Mg atoms were enriched at the end of the phases near the dendrites. The distribution of Zn was relatively uniform in the grains after 400After homogenized, the color of Cu, Mg and Zn in the grains became more uniform as Figure 4c, f and i. No dark color area was found in grains. The two-step homogenization treatment successfully eliminated the segregation of Cu, Mg and Zn. "

"Note that the color of Cu, Mg and Zn in the grains changed during the homogenization process. The color of Al matrix became light after homogenization, which displayed blue in Figure 4c of Cu and green in Figure 4f of Mg. This result indicated a rise of Cu and Mg content in the matrix. In contrast, the concentration of Zn in the grains decreased in Figure 4i. The results revealed that Cu and Mg atoms tended to move from the coarse phases on the dendrites towards the Al matrix during homogenization treatment, while the movement of Zn atoms was opposite, from the Al matrix towards the second phases. "

Point 4:  

Thank you for your comment. The wrong sub-title has been revised as "3.3 constituent particles" (line 177).

Point 5:  Response 5: 

Point 6:  

Thank you for your comments. According to your comments, we also find that the previous writing was incorrect. The revised statements are as follows:

Section 3.4: (line 220-223)

"Precipitation of the alloy subjected to two-step homogenization was detected by HRTEM and STEM-HADDF in Figure 7, where the zone axis of diffraction pattern was along [110]Al in Figure 7a. The diffraction pattern showed the spots distributed symmetrically at                                                and their equivalent positions, which was represented for Al3Zr in Al-Li alloy with the zone axis of [110]Al . "

Section 3.4: (line 242-244)

"Figure 7. TEM images on disperoids with (a) diffraction pattern with zone axis of [110]Al through (b) bright field and (c) dark field, (d) STEM-HAADF and elements mapping of (e) Al and (f) Zr in the particle and (g) HRTEM image of the fine precipitates with FFT. "

Point 7: "The Al3Zr primary phase does not meet this characteristic due to its short number and large size." What does short number mean? Also, are you referring to 20-30 nm precipitates as large particles? Primary Al3Zr precipitates, formed during solidification, can be a few microns across and visible with an optical microscope. Another issue is the fact that this is a homogenized alloy with the highest homogenization temperature being 500℃. 

Response 7: . The main reason we use "preferential precipitates" is to distinguish the Al3Zr particles from the tiny Al3(ZrxTiyLi1-x-y) particles. Their spots in diffraction pattern are the same. We feel very sorry that we did not express our experimental results clearly. We rewrote the section 3.4 as follows:

Section 3.4: (line 220-244)

"Precipitation of the alloy subjected to two-step homogenization was detected by HRTEM and STEM-HADDF in Figure 7, where the zone axis of diffraction pattern was along [110]Al in Figure 7a. The diffraction pattern showed the spots distributed symmetrically at  and their equivalent positions, which was represented for Al3Zr in Al-Li alloy with the zone axis of [110]Al. A few spherical phases with size of 20-30nm were observed in the grain in Figure 7b, which were coherent with the matrix. According to the dark field image in Figure 7c, in addition to the large spherical particles, a large number of tiny bright spots were also observed in the yellow square. Element analysis in the spherical phase with size of 20-30nm was carried out by HADDF-Mapping in Figure 6d, e and f, which provided the experimental evidence of Zr atoms enriched in the spherical phase. In summary, the spherical phase was judged as a coherent phase Al3Zr with FCC-based L12 structure formed during homogenization at 400 preferentially [23]. However, it is worth noting that although the diffraction patterns of Al3Zr was very bright and sharp at the center, the outer part was weak and the light was divergent, which is a typical characteristic of the diffraction spots formed by plenty of small-sized precipitates. The Al3Zr phase with size of ~20nm does not meet with this characteristic. The HRTEM was detected in Figure 7g, showing the tiny phases all over the matrix with size of 1-2nm pointed as yellow arrows, which might be the tiny bright spots in Figure 7c. These nanoscale precipitates were consistent with the characteristic of diffraction spots, presumably as tiny Al3(ZrxTiyLi1-x-y) particles. These uniform and tiny particles provided the alloy better thermal stability due to the strong pinning dislocation ability, which is a very rare phenomenon in alloys microalloying no Mg or Zn. The large number of Mg-Zn atom clusters diffused into the matrix during homogenization may be the key to forming Al3(ZrxTiyLi1-x-y) particles. "

Figure 7. TEM images on disperoids with (a) diffraction pattern with zone axis of [110]Al through (b) bright field and (c) dark field, (d) STEM-HAADF and elements mapping of (e) Al and (f) Zr in the particle and (g) HRTEM image of the fine precipitates with FFT.

Use of English must be improved. Please work with a professional language editor.

Response 8:  

"Microstructure characterizations before and after homogenization of the samples were carried out by using Leica DMI3000 polarizing microscope. The samples were subjected to electrolytic polishing by HBF3 (16.8g/L). "

Section 3.1: (line 116-117)

"The elements distribution of Cu, Mg and Zn was investigated by EPMA in the as-cast, first-step homogenized (400, 8h+500, 24h) alloys in Figure 4, where the concentration of alloying elements in different regions was listed next to the Figure. The non-uniform distribution of Cu, Mg and Zn was obvious in the as-cast alloy. "

Section 3.2 (line 130-133)

"The Mg and Zn tended to be concentrated together on and around the dendrites. After the first-step homogenization, the color of Cu in the coarse continuous phases had merely changed comparatively in Figure 4a and b, which indicated that the diffusion of Cu was not clear at 400. "

Section 3.2 (line 136-140)

"The distribution of Zn was relatively uniform in the grains after 400After homogenized, the color of Cu, Mg and Zn in the grains became more uniform as Figure 4c, f and i. No dark color area was found in grains. The two-step homogenization treatment successfully eliminated the segregation of Cu, Mg and Zn. "

Section 3.2 (line 141-147)

"Note that the color of Cu, Mg and Zn in the grains changed during the homogenization process. The color of Al matrix became light after homogenization, which displayed blue in Figure 4c of Cu and green in Figure 4f of Mg. This result indicated a rise of Cu and Mg content in the matrix. In contrast, the concentration of Zn in the grains decreased in Figure 4i. The results revealed that Cu and Mg atoms tended to move from the coarse phases on the dendrites towards the Al matrix during homogenization treatment, while the movement of Zn atoms was opposite, from the Al matrix towards the second phases. "

"No enrichment of Zr, Mn and Fe were observed at area A. "

Section 3.4 (line 225,226)

"According to the dark field image in Figure 7c, in addition to the large spherical particles, a large number of tiny bright spots were also observed in the yellow square. "

Section 3.4 (line 233-235)

"The Al3Zr phase with size of ~20nm does not meet with this characteristic. The HRTEM was detected in Figure 7g, showing the tiny phases all over the matrix with size of 1-2nm pointed as yellow arrows, which might be the tiny bright spots in Figure 7c. "

We appreciate for Editors/Reviewers’ warm work earnestly, and hope that the correction will meet with approval.  

Round  2

Reviewer 1 Report

The authors seem to amend the manuscript more or less as I requested. The last comment, after which the paper can be proceed for publication:

- There is a bit of discussion about the evolution of grain size. There are etching techniques and optical microscopy with polarized light, as somehow be used in the manuscript. But one of the most reliable technique is EBSD. Even if authors can not do this analysis, it should be mentioned in the text. An example of using this technique for cast Al alloys that can be referred to: https://doi.org/10.1016/j.msea.2017.07.074

Author Response

Response to Reviewer 1 Comments

Thank you for the reviewer comments concerning our manuscript entitled “Investigation on microstructural evolution and properties of an Al-Cu-Li alloy with Mg and Zn microalloying during homogenization” (manuscript ID: metals-387124). Those comments are all valuable and helpful for revising and improving our paper. We have studied comments carefully and have made correction which we hope meet with approval. Revised portions are highlighted in yellow in the revised manuscript. The responds to the reviewer’s comments as well as main corrections in the paper are as follows. In addition, the lines of the revised statements in the revised manuscript are shown in brackets.

Point 1: There is a bit of discussion about the evolution of grain size. There are etching techniques and optical microscopy with polarized light, as somehow be used in the manuscript. But one of the most reliable technique is EBSD. Even if authors can not do this analysis, it should be mentioned in the text. An example of using this technique for cast Al alloys that can be referred to: https://doi.org/10.1016/j.msea.2017.07.074

Response 1: Thank you for your comments. Your comments are very helpful and we have studied the reference carefully. EBSD is indeed a very effective and reliable method when detecting the grain size of cast Al alloys. According to your comments, the statements about EBSD and grain size have been mentioned in the revised manuscript as follows:

Section 3.1: (line 116-119)

"There existed typical dendrites in the as-cast alloy in Figure 3a obviously. Precise grain size can be obtained by EBSD according to the previous work published by Martin Riestra [21]. In the optical micrograph, the average size of grains was estimated roughly to be 350-450μm. "

References: (line 430-432)

21. Riestra, M.; Ghassemali, E.; Bogdanoff, T.; Seifeddine, S. Interactive effects of grain refinement, eutectic modification and solidification rate on tensile properties of Al-10Si alloy, Mater. Sci. Eng. A. 2017, 703, 270–279.

We appreciate for Editors/Reviewers’ warm work earnestly, and hope that the correction will meet with approval. Once again, thank you very much for your comments and suggestions.

Corresponding author: Hongying Li

Reviewer 2 Report

I appreciate the revisons of the article done by the authors.

Author Response

Response to Reviewer 2 Comments

Thank you for the reviewer comments concerning our manuscript entitled “Investigation on microstructural evolution and properties of an Al-Cu-Li alloy with Mg and Zn microalloying during homogenization” (manuscript ID: metals-387124). Those comments are all valuable and helpful for revising and improving our paper. The responds to the reviewer’s comments are as follows.

Point 1: I appreciate the revisons of the article done by the authors.

Response 1: Thank you very much. Your approval is greatly appreciated and we will continue to improve the paper.

We appreciate for Editors/Reviewers’ warm work earnestly, and hope that the correction will meet with approval. Once again, thank you very much for your comments and suggestions.

Corresponding author: Hongying Li

E-mail:[email protected]
